

# LiSBOA: LiDAR Statistical Barnes Objective Analysis for optimal design of LiDAR scans and retrieval of wind statistics. Part II: Applications to synthetic and real LiDAR data of wind turbine wakes

Stefano Letizia, Lu Zhan, and Giacomo Valerio Iungo

Wind Fluids and Experiments (WindFluX) Laboratory, Mechanical Engineering Department, The University of Texas at Dallas, 800 W Campbell Rd, 75080 Richardson, TX, USA

**Correspondence:** Giacomo Valerio Iungo, (valerio.iungo@utdallas.edu)

**Abstract.** The LiDAR Statistical Barnes Objective Analysis (LiSBOA), presented in Letizia et al., is a procedure for the optimal design of LiDAR scans and calculation over a Cartesian grid of the statistical moments of the velocity field. The LiSBOA is applied to LiDAR data collected in the wake of wind turbines to reconstruct mean and turbulence intensity of the wind velocity field. The proposed procedure is firstly tested for a numerical dataset obtained by means of the virtual LiDAR technique applied to the data obtained from a large eddy simulation (LES). The optimal sampling parameters for a scanning Doppler pulsed wind LiDAR are retrieved from the LiSBOA, then the estimated statistics are calculated showing a maximum error of about 4% for both the normalized mean velocity and the turbulence intensity. Subsequently, LiDAR data collected during a field campaign conducted at a wind farm in complex terrain are analyzed through the LiSBOA for two different configurations. In the first case, the wake velocity fields of four utility-scale turbines are reconstructed on a 3D grid, showing the capability of the LiSBOA to capture complex flow features, such as high-speed jet around the nacelle and the wake turbulent shear layers. For the second case, the statistics of the wakes generated by four interacting turbines are calculated over a 2D Cartesian grid and compared to the measurements provided by the nacelle-mounted anemometers. Maximum discrepancies as low as 3% for the normalized mean velocity and turbulence intensity endorse the application of the LiSBOA for LiDAR-based wind resource assessment and diagnostic surveys for wind farms.



## 15 List of LiSBOA symbols

- $L$: number of realizations/scans

- $\Delta\theta$: angular resolution

- $\sigma$: smoothing parameter

- $m$: number of iterations

- $R_{\mathrm{max}}$: radius of influence

- $\boldsymbol{\Delta n}$: half-wavelength

- $\boldsymbol{\Delta n_0}$: fundamental half-wavelength

- $\Delta d$: random data spacing

- $\boldsymbol{dx}$: resolution of the Cartesian grid

- $D^m$: response at the $m$-th iteration

- $\epsilon^I$: cost function I (data loss)

- $\epsilon^{II}$: cost function II (standard deviation of the sample mean)

- $\tau$: integral time-scale

- $\tau_a$: accumulation time

- $\Delta r$: gate length

- $N_r$: number of points per beam

- $T$: total sampling time

- $\tilde{\cdot}$: spatial variable in the scaled frame of reference



# 1 Introduction

The use of Doppler light detection and ranging (LiDAR) technology for wind energy applications has largely increased over the last decade (Clifton et al., 2018; Veers et al., 2019). Thanks to the achieved measurement accuracy, simpler and cost-effective deployments compared to traditional met-tower instrumentation, this remote sensing technique is now included in the international standards as a reliable tool for performance diagnostic of wind turbines and wind resource assessment (International Electrotechnical Commission 61400-12-1, 2017). Nonetheless, due to the limited spatio-temporal resolution and the distribu-

tion of the sample points in a spherical reference frame, the reconstruction of wind statistics from LiDAR samples still presents several challenges (Sathe et al., 2011; Newman et al., 2016).

In the companion paper (Letizia et al.), we presented a revisited Barnes objective analysis (Barnes, 1964) for the calculation of wind statistics from scattered LiDAR data, which is referred to as LiDAR Statistical Barnes Objective Analysis (LiSBOA). This procedure enables the estimation over a Cartesian grid of the mean, variance and even higher-order central statistical

moments of the radial velocity field probed by a scanning Doppler pulsed wind LiDAR. The LiSBOA performs also adequate filtering of small-scale variability in the mean field and mitigation of the dispersive stresses on the higher-order statistics provided that the algorithm is tuned based on the characteristics of the flow under investigation and the data collection strategy is optimally designed through the LiSBOA.

The LiSBOA capability to estimate statistics of an ergodic turbulent velocity field makes it a suitable tool for the analysis

of wind turbine wakes and the resource assessment of sites characterized by heterogeneous wind conditions, such as in presence of flow distortions induced by a complex terrain. Over the last decade, wind LiDARs have been used to investigate wind turbine wakes; for instance, Käsler et al. (2010) and Clive et al. (2011) measured the velocity deficit past utility-scale wind turbines, while Bingöl et al. (2010) used a nacelle-mounted LiDAR to detect wake displacements and validate the dynamic wake meandering model (Larsen et al., 2008). Fitting of the wake velocity deficit was successfully exploited to extract quan-

titative information about wake evolution from LiDAR measurements (Aitken and Lundquist, 2014; Wang and Barthelmie, 2015; Kumer et al., 2015; Trujillo et al., 2016; Bodini et al., 2017).

A deeper understanding on the physics of turbine wakes was achieved by calculating temporal (Trujillo et al., 2011; Iungo et al., 2013b; Iungo and Porté-Agel, 2014; Kumer et al., 2015; Machefaux et al., 2016; Van Dooren et al., 2016) or conditional (Aubrun et al., 2016; Machefaux et al., 2016; Garcia et al., 2017; Bromm et al., 2018; Iungo et al., 2018; Zhan et al., 2019,

2020) statistics of the velocity collected through LiDAR scans performed at different times. Using this approach, Iungo and Porté-Agel (2014) detected a significant dependence of the wake recovery rate on atmospheric stability based on time-averaged volumetric LiDAR scans. The same concept was expanded by other authors using ensemble statistics (Machefaux et al., 2016; Carbajo Fuertes et al., 2018; Zhan et al., 2019, 2020). Kumer et al. (2015) carried out a comparison between instantaneous, 10 minutes and daily-averaged velocity and turbulence intensity fields around utility-scale wind turbines, highlighting the

presence of persistent turbulent wakes. Trujillo et al. (2011) used a nacelle-mounted LiDAR to quantify meandering-induced wake diffusion and added turbulence from statistics calculated over 10-minute periods.





Second-order statistics are of great interest in wind energy. Iungo et al. (2013b) used velocity time-series extracted from LiDAR fixed scans performed downstream of a 2-MW wind turbine to detect enhanced turbulence intensity in the proximity of the wake shear layers. More recently, temporal statistics over 30-minute periods allowed for the identification of turbulent

wake shear layers from both numerical (Fuertes Carbajo and Porté-Agel, 2018) and experimental (Carbajo Fuertes et al., 2018) velocity fields. Aubrun et al. (2016) attempted to characterize the turbulence intensity using bin statistics, even though achieving values higher than expected, i.e. larger than 50%. Zhan et al. (2019) used clustered data of wake velocity fields to retrieve a proxy for the standard deviation of wind speed in the wake of utility-scale turbines. These authors reported significant variability in the wake turbulent statistics depending on the atmospheric stability regime and operative conditions of the wind

turbines.

For the above-mentioned technical features of LiDARs, these remote sensing instruments are now also used for wind resource assessment (Liu et al., 2019) enabling estimates of wind statistics for broad ranges of wind conditions and site typology, such as for flat terrains (Karagali, 2018; Sommerfeld et al., 2019; Sanchez-Gomez and Lundquist, 2019), complex terrains (Krishnamurthy et al., 2011, 2013; Pauscher et al., 2016; Kim et al., 2016; Vasiljević et al., 2017; Karagali, 2018; Risan et al.,

2018; Menke et al., 2019; Fernando et al., 2019), near-shore (Hsuan et al., 2014; Floors et al., 2016; Shimada et al., 2018) and off-shore locations (Pichugina et al., 2012; Koch et al., 2014; Gottschall, 2018; Viselli et al., 2019). LiDAR scanning strategies for wind resource assessment encompass Doppler beam swinging (DBS) (Hsuan et al., 2014; Pauscher et al., 2016; Kim et al., 2016; Shimada et al., 2018; Gottschall, 2018; Viselli et al., 2019; Sommerfeld et al., 2019; Sanchez-Gomez and Lundquist, 2019), Plan Position Indicator (PPI) scans (Krishnamurthy et al., 2011, 2013; Pauscher et al., 2016; Floors et al.,

2016; Vasiljević et al., 2017; Karagali, 2018), Range Height Indicator (RHI) scans (Pichugina et al., 2012; Floors et al., 2016; Menke et al., 2019; Fernando et al., 2019) or fixed scans (Risan et al., 2018). Statistics are generally calculated based on the canonical 10-minute periods assuming steady inflow conditions, while linear interpolation is widely used for data post-processing.

In the light of great relevance for the wind energy applications of the statistical analysis of wind LiDAR data, for this work,

the LiSBOA procedure is applied to virtual and real LiDAR measurements of wind turbine wakes. The scope of this study is dual: first, assessing the capabilities provided by the LiSBOA for the optimal design of the LiDAR scanning strategy by maximizing the statistical accuracy of the measurements and coverage of the sampling domain with the prescribed spatial resolution; second, showing the potential of the LiSBOA to reconstruct mean velocity and turbulence intensity fields from LiDAR data to unveil important flow features of wind turbine wakes.

With these aims, the LiSBOA is initially applied to virtual LiDAR data generated by a LiDAR simulator scanning the wake of a turbine modeled through the actuator disk approach in LES environment. This numerical test case allows for an extensive error analysis enabling the quantification of the LiSBOA accuracy. Then, real LiDAR data collected in the wakes generated by four 1.5-MW wind turbines are analyzed through the LiSBOA. Specific wake features, such as the high-speed jet around the nacelle and the turbulent shear layers, as well as perturbations induced by the complex topography, are detected. Finally,

to provide a quantitative comparison with the data retrieved by means of traditional anemometers, the LiSBOA is employed to calculate mean velocity and turbulence intensity fields of the wakes generated by four 1-MW turbines interacting each other.





The remainder of the manuscript is organized as follows: Sect. 2 reports the results of the virtual LiDAR data analysis, along with the description of the LiDAR simulator. Section 3.3.1 provides a description of the site and the experimental setup of the field campaign. In Sect. 3.3.2, the scan design and the reconstruction of the statistics of the non-interacting wakes are discussed, while Sect. 3.3.3 presents the results of the comparison between nacelle anemometer statistics and LiSBOA for the multiple interacting wakes. Finally, conclusions are drawn in Sect. 4. The paper uses symbols introduced in the companion paper Letizia et al., which the reader is encouraged to review for a better understanding of the present manuscript.

## 2  LiSBOA validation against virtual LiDAR data

The LiSBOA algorithm is applied to a synthetic dataset generated through the virtual LiDAR technique to assess accuracy in the calculation of statistics for a turbulent flow. For this purpose, a simulator of a scanning Doppler pulsed wind LiDAR is implemented to extract the line-of-sight velocity from a numerical velocity field produced through high-fidelity large-eddy simulations (LES). Due to their simplicity and low computational costs, LiDAR simulators have been widely used for the assessment of post-processing algorithms of LiDAR data and scan design procedures (Mann et al., 2010; Stawiarski et al., 2015; Lundquist et al., 2015; Mirocha et al., 2015).

A main limitation of LiDARs is represented by the spatio-temporal averaging of the velocity field, which is connected with the acquisition process. Three different types of smoothing mechanisms can occur during the LiDAR sampling: the first is the averaging along the laser beam direction within each range gate, which has been commonly modeled through the convolution of the actual velocity field with a weighting function within the measurement volume (Smalikho, 1995; Frehlich, 1997; Sathe et al., 2011). The second process is the time-averaging associated with the sampling period required to achieve a back-scattered signal with adequate intensity (O'Connor et al., 2010; Sathe et al., 2011), while the last one is the transverse averaging (azimuth-wise or elevation-wise) occurring in case of a scanning LiDAR operating in continuous mode (Stawiarski et al., 2013). These filtering processes lead to significant underestimation of the turbulence intensity (Sathe et al., 2011), overestimation of integral length-scales (Stawiarski et al., 2015), and damping of energy spectra for increasing wavenumbers (Risan et al., 2018; Puccioni and Iungo, 2020).

Three versions of a LiDAR simulator are implemented for this work: the simplest one is referred to as ideal LiDAR, which samples the LES velocity field at the experimental points through a nearest-neighbor interpolation. This method minimizes the turbulence damping while retaining the geometry of the scan and the projection of the velocity onto the laser beam direction. The second version of the LiDAR simulator reproduces a step-stare LiDAR, namely the LiDAR scans for the entire duration of the accumulation time at a fixed direction of the LiDAR laser beam. Two filtering processes take place for this configuration: beam-wise convolution and time averaging. To model the beam-wise average, the retrieval process of the Doppler LiDAR is reproduced by means of a spatial convolution (Mann et al., 2010):

$$u_{\mathrm{LOS}}(\boldsymbol{x},t) = \int\limits_{-\infty}^{\infty} \phi(s)\boldsymbol{n}\cdot\boldsymbol{u}(\boldsymbol{x}+\boldsymbol{n}s,t)ds, \tag{1}$$




where $\boldsymbol{n}$ is the LiDAR laser-beam direction. A triangular weighting function $\phi(s)$ was proposed by Mann et al. (2010):

$$
\phi(s) = \begin{cases} \frac{\Delta r/2 - |s|}{\Delta r^2/4} & \text{if } |s| < \Delta r/2 \\ 0 & \text{otherwise,} \end{cases} \tag{2}
$$

where $\Delta r$ is the gate length. The former expression is valid assuming matching time windowing, i.e. gate length equal to the pulse width, and the velocity value is retrieved based on the first momentum of the back-scattering spectrum. Despite its simplicity, Eq. (2) has shown to estimate realistic turbulence attenuation due to the beam-wise averaging process of a pulsed Doppler wind LiDAR (Mann et al., 2009). Furthermore, a time-averaging occurs due to the accumulation time necessary for the LiDAR to acquire a velocity signal with sufficient intensity and, thus, signal-to-noise-ratio. This process is modeled through

a window average within the acquisition interval of each beam. For the sampling of the LES velocity field in space and time, a nearest-neighbor interpolation method is used.

The third version of the LiDAR simulator mimics a pulsed LiDAR operating in continuous mode and performing PPI scans, where, in addition to the beam-wise convolution and time-averaging, azimuth-wise averaging occurs due to the variation of the LiDAR azimuth angle of the scanning head during the scan. The latter is taken into account by adding to the time average an

azimuthal averaging among all data points included within the following angular sector:

$$
\begin{cases} |\theta - \theta_p| < \Delta\theta/2 \\ |\beta - \beta_p| < \sin^{-1}\left(\frac{\Delta z}{2r_p}\right) \end{cases} \tag{3}
$$

where $r$ is the radial distance from the emitter, while $\theta$ and $\beta$ are the associated azimuth and elevation angles, respectively. The subscript $p$ refers to the $p$-th LiDAR data point. Following the suggestions by Stawiarski et al. (2013), the out-of-plane thickness, $\Delta z$, is considered equal to the length of the diagonal of a cell of the computational grid.

As a case study, we use the LES dataset of the flow past a single turbine with the same characteristics of the 5-MW NREL reference wind turbine (Jonkman et al., 2009). The rotor is three-bladed and has a diameter $D = 126$ m. The tip-speed-ratio of the turbine is set to its optimal value of 7.5. A uniform incoming wind with freestream velocity of $U_\infty = 10$ m s$^{-1}$ and turbulence intensity of $3.6\%$ is considered. The rotor is simulated through an actuator disk with rotation, while the drag of the nacelle is taken into account using an immersed boundary method (Ciri et al., 2017). More details on the LES solver

can be found in Santoni et al. (2015). The computational domain has dimensions $(L_x \times L_y \times L_z = 9D \times 6D \times 10D)$ in the streamwise, spanwise, and vertical directions, respectively, and it is discretized with $512 \times 256 \times 384$ grid points, respectively. A radiative condition is imposed at the outlet, freeslip is enforced on the top and bottom while periodicity is applied in the spanwise direction. Ergodic velocity vector fields are available for a total time of $T = 750$ s. Figure 1a shows a snapshot of the streamwise velocity field over the horizontal plane at hub height obtained from the LES. The respective data of the radial

velocity obtained from the three versions of the LiDAR simulator by considering a scanning pulsed wind LiDAR deployed at the turbine location and at hub height highlight the increased spatial smoothing of the radial velocity field by adding the various averaging processes connected with the LiDAR measuring process, namely beam-wise, temporal and azimuthal averaging (Fig. 1).

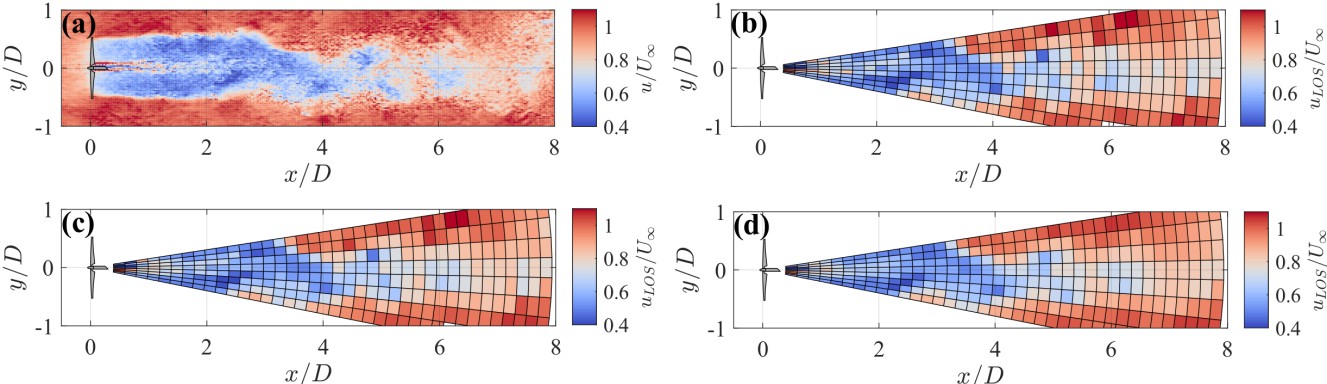

**Figure 1.** Snapshot at the hub-height horizontal plane of the wake generated by the 5-MW NREL reference wind turbine: **(a)** LES streamwise velocity; **(b)** ideal virtual LiDAR with angular resolution $\Delta\theta = 2.5°$ and zero elevation; **(c)** step-stare virtual LiDAR (same settings); **(d)** continuous mode virtual LiDAR (same settings).

The optimal design of the virtual LiDAR scan is based on the procedure outlined in Letizia et al. (see specifically the flowchart provided in Fig. 7). For the estimation of the flow characteristics, the azimuthally-averaged mean and standard deviation of streamwise velocity, as well as the integral time-scale are considered (Fig. 2). The use of cylindrical coordinates is justified by the axisymmetry of the statistics of the wake velocity field generated by a turbine operating in a uniform velocity field (Iungo et al., 2013a; Viola et al., 2014; Ashton et al., 2016).

The streamwise LES velocity field shows the presence of a higher-velocity jet surrounding the nacelle, while $\overline{u}/U_\infty$ exhibits a clear minimum placed at $y/D \sim 0.25$ (Fig. 2a). These flow features are consistent with the double-Gaussian velocity profile typically observed at the near-wake region (Aitken and Lundquist, 2014). In Fig. 2b, the standard deviation of the streamwise velocity has high values in the very near-wake ($x/D < 1$) in the proximity of the rotor axis, which is most probably connected with the vorticity structures generated in proximity of the rotor hub and their dynamics (Iungo et al., 2013a; Viola et al., 2014; Ashton et al., 2016). Similarly, enhanced values of the velocity standard deviation occur at the wake boundary ($r/d \approx 0.5$), which are connected with the formation and dynamics of the helicoidal tip vortices (Ivanell et al., 2010; Debnath et al., 2017). A peak of $\sqrt{\overline{u'^2}}/U_\infty$ is observed around ($x/D \approx 3$), which can be considered as the formation length of the tip vortices. The integral time-scale is evaluated integrating the sample biased autocorrelation function up to the first zero-crossing (Zieba and Ramza, 2011). The integral time-scale is generally smaller within the wake than for the typical values observed in the freestream, which is consistent with the smaller dimensions of the wake vorticity structures than for larger energy-containing structure present in the incoming turbulent wind.

To reconstruct the mentioned flow features, the fundamental half-wavelengths in the spanwise and vertical directions selected for this application of the LiSBOA are $\Delta n_{0,y} = \Delta n_{0,z} = 0.5D$. Furthermore, considering the streamwise elongation of the isocontours of the flow statistics shown in Fig. 2a, a conservative value of the fundamental half-wavelength in the $x$-direction $\Delta n_{0,x} = 2.5D$ is selected. The relevance of the selected $\boldsymbol{\Delta n_0}$ is tested by evaluating the 3D energy spectrum of $\overline{u}/U_\infty$ and





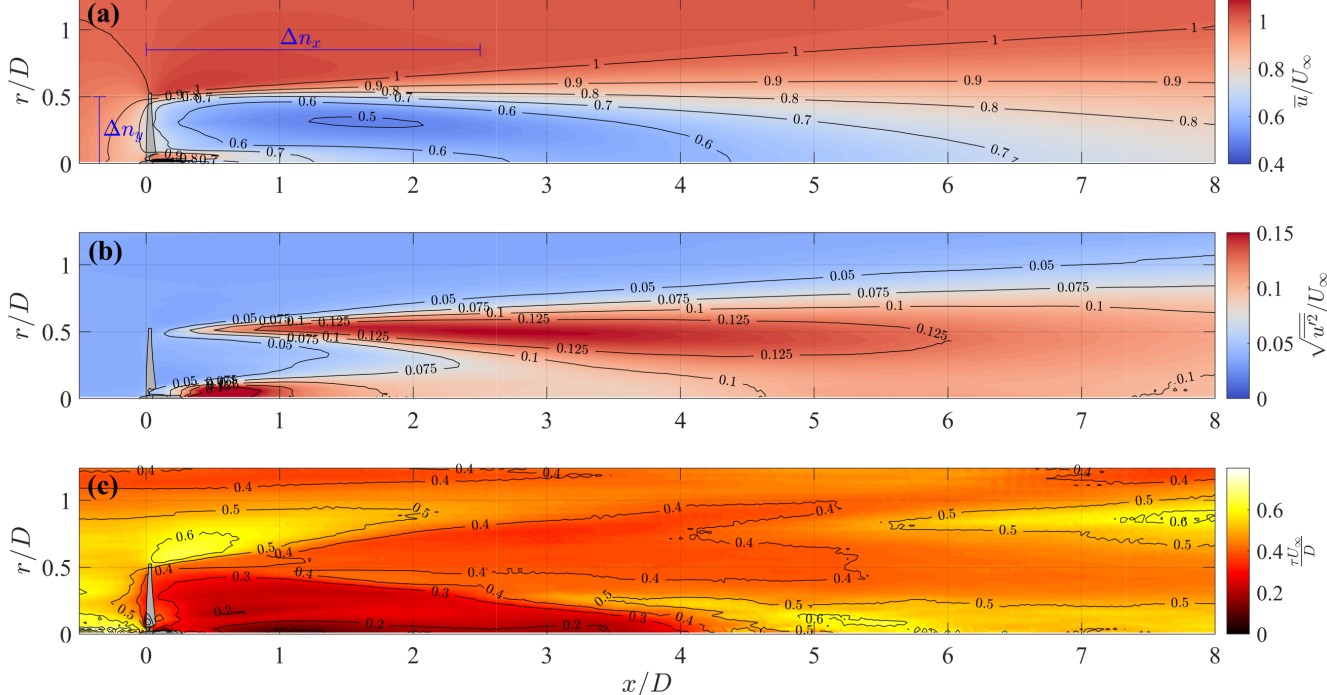

**Figure 2.** Azimuthally-averaged statistics of the LES streamwise velocity field: **(a)** mean value; **(b)** standard deviation; **(c)** integral time scale.

$\overline{u'^2}/U_\infty^2$ in the physical and in the scaled reference frame (see Letizia et al., Eq. (6)). The spectra are azimuthally averaged by exploiting the axisymmetry of the wake. The spectra in the physical reference frame (Figs. 3a and b) reveal the clear signature of a streamwise elongation of the energy-containing scales for both velocity mean and variance, with the energy being spread over a larger range of frequencies in the radial direction compared to the streamwise direction. After the scaling (Figs. 3c and d), the spectra become more isotropic in the spectral domain, namely the energy is distributed equally along the $\tilde{k}_x$ and $\tilde{k}_r$ axes. In Fig. 3c, the blue dashed line represents the intersection with the $\tilde{k}_x$ - $\tilde{k}_r$ plane of the spherical isosurface that in the wavenumber space is characterize by $D^m(\Delta \tilde{n}_0) = 0.95$. Based on the procedure defined in Letizia et al., all the modes contained within that sphere are reconstructed with a response $D^m > 0.95$, while higher-frequency features lying outside will be damped. Numerical integration of the 3D energy spectrum shows that 94% of the total spatial variance of the mean is contained within that sphere, which ensures that the energy-containing modes in the mean flow are adequately reconstructed with the selected parameters.

In Fig. 2, the statistics guide for the selection of the parameters necessary to estimate the standard deviation of the sample mean, $\epsilon^{II}$ (see Eq. (14) of Letizia et al.). Specifically, $\sqrt{\overline{u'^2}}/U_\infty = 0.1$ and $\tau U_\infty/D = 0.4$ ($\tau \sim 5$ s) are deemed representative of the wake region.

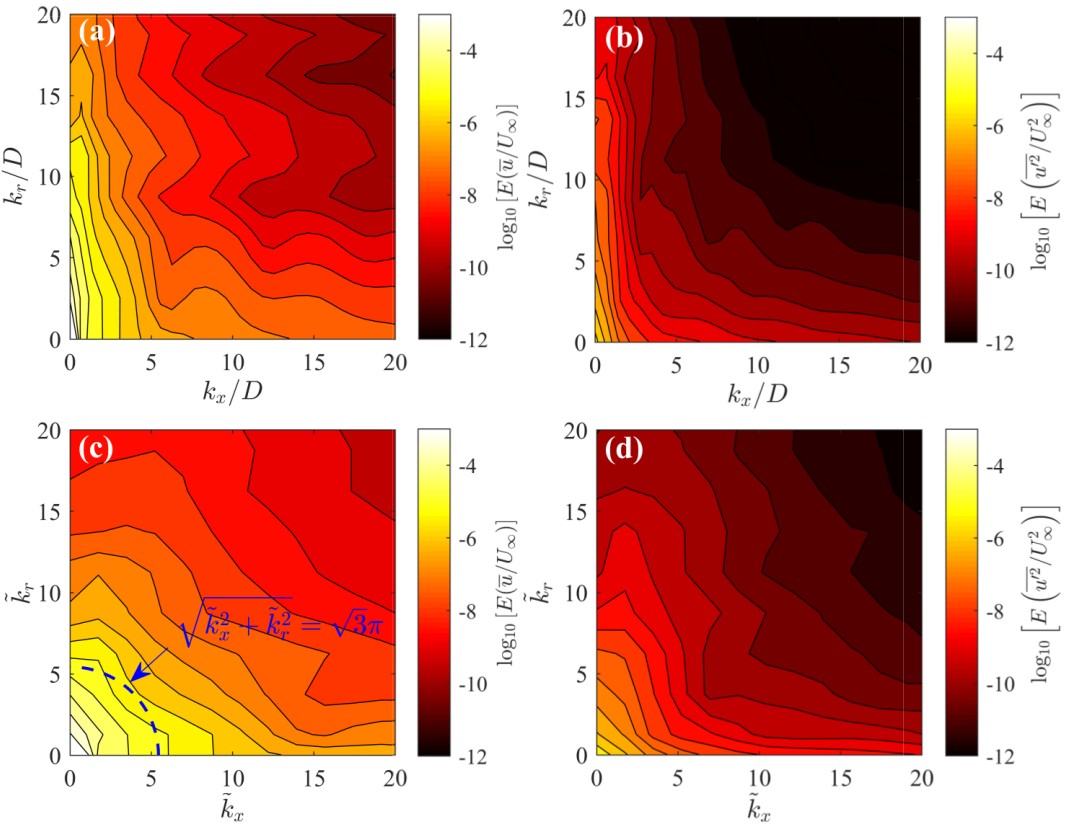

**Figure 3.** Azimuthally-averaged energy spectra of the LES velocity fields : **(a)** mean streamwise velocity on the physical domain; **(b)** variance of streamwise velocity on the physical domain; **(c)** mean streamwise velocity on the scaled domain; **(d)** variance of streamwise velocity on the scaled domain. The blue dashed line indicates wavenumbers reconstructed with response equal to $D^m(\mathbf{\Delta\tilde{n}_0}) = 0.95$.

The application of the LiSBOA also requires to provide technical specifications of the LiDAR, specifically accumulation
time, $\tau_a$, number of gates, $N_r$, and gate length, $\Delta r$. For this work, these parameters are selected based on the typical settings of
LiDARs Windcube 200S and StreamLine XR (El-Asha et al., 2017; Zhan et al., 2019, 2020), namely $\tau_a = 0.5$ s, $N_r = 39$ and
$\Delta r = 25$ m. Furthermore, to probe the wake region, a volumetric scan including several PPI scans with azimuth and elevation
angles uniformly spanning the range $\pm 10°$ with a constant angular resolution, $\Delta\theta$, is selected, while the virtual LiDAR is
placed at the turbine hub.

With the information provided about the flow under investigation and the LiDAR system, it is possible to draw the Pareto
front for the optimization of the LiDAR scan as a function of the angular resolution of the LiDAR scanning head, $\Delta\theta$, and the
smoothing parameter of the LiSBOA, $\sigma$, as shown in Fig. 4 for the case under investigation. As described in Letizia et al., the
optimization problem of a LiDAR scan consists of minimizing two cost functions. The first cost function, $\epsilon^I$, represents the
percentage of grid nodes for which the Petersen-Middleton constraint applied to the smallest half-wavelength of interest, $\mathbf{\Delta n_0}$,





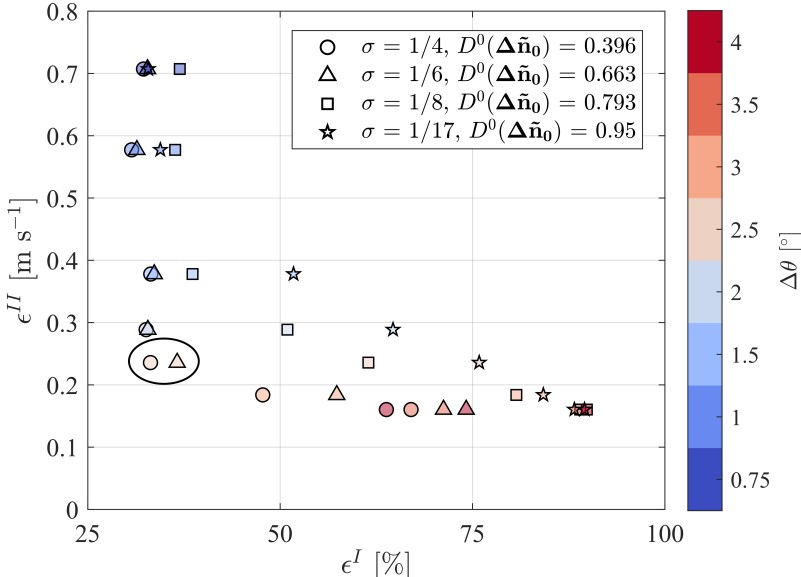

**Figure 4.** Pareto front for the design of the optimal LiDAR scan for the LES dataset. The circle indicates the selected optimal configurations.

is not satisfied. In other words, it represents the percentage of the measurement domain that is not sampled with adequate data spacing so that aliasing of the LiDAR data may occur. The second cost function, $\epsilon^{II}$, is the standard deviation of the sample mean and quantifies the temporal statistical uncertainty due to the finite number of scan repetitions, $L$, allowed within the total sampling time, $T$.

For the optimization of the LiDAR scan, the LiDAR angular resolution, $\Delta\theta$, is evenly varied, for a total number of 7 cases, from $0.75°$, corresponding to a total number of scans $L = 2$, to $4°$, which corresponds to $L = 41$. The four values of $\sigma$ recommended in Table 2 of Letizia et al. to achieve a response of the mean $D^m(\Delta\tilde{n}_0) = 0.95$ are considered here. In Fig. 4, markers indicate the different $\sigma$ and, thus, the response of high-order statistical moments, $D^0(\Delta\tilde{n}_0)$. The Pareto front shows that increasing $\Delta\theta$ from $0.75°$ up to $2.5°$ drastically reduces the uncertainty on the mean ($\epsilon^{II}$) by roughly 70 %, but does not affect significantly data loss consequent to the enforcement of the Petersen-Middleton constraint ($\epsilon^I$). For larger angular resolutions, the statistical significance improves just marginally, but at the cost of a relevant data loss. For $\Delta\theta > 2.5°$, in particular, $\epsilon^I$ becomes extremely sensitive to $\sigma$, with the most severe data loss occurring for small $\sigma$ (i.e. small $R_{max}$). The Pareto front also shows that to achieve a higher response for the higher-order statistics, $D^0(\Delta\tilde{n}_0)$, generally entails an increased data loss and/or statistical uncertainty of the mean. This analysis suggests that the optimal LiDAR scan for the reconstruction of the mean velocity field should be performed with $\Delta\theta = 2.5°$ and $\sigma = 1/4$ or $1/6$.

Virtual LiDAR simulations are performed for all the values of angular resolution utilized in the Pareto front reported in Fig. 4. The streamwise component is estimated from the line-of-sight velocity through an equivalent velocity approach (Zhan et al., 2019), then mean velocity and turbulence intensity (i.e. the ratio between standard deviation and mean) are reconstructed





through the LiSBOA. The maximum error is quantified through the 95-th percentile of the absolute error, $AE_{95}$, using as reference the LES statistics interpolated on the LiSBOA grid.

Fig. 5 reports the $AE_{95}$ for the flow statistics for all the virtual experiments. The error for the mean field (Figs. 5a-c) is mostly governed by the angular resolution, with a higher error occurring for slower scans. This is a clear consequence of the increased statistical uncertainty due to the limited number of scan repetitions, $L$, that are achievable for small $\Delta\theta$ values and a fixed total sampling period, $T$, while $AE_{95}$ stabilizes for $\Delta\theta \geq 2.5°$. The trend of the $AE_{95}$ for $\overline{u}/U_\infty$ with the pair smoothing parameter-number of iterations, $\sigma - m$, is less significant since the theoretical response of the fundamental mode is ideally

equal for all the four cases. Conversely, the error on the turbulence intensity (Figs. 5d-f) shows low sensitivity to the angular resolution but a steep increase for small $\sigma$ values, which is due to the reduction of the radius of influence, $R_{\mathrm{max}}$, and number of points averaged per grid node.

From a more technical standpoint, the error on the mean velocity field, $\overline{u}/U_\infty$, appears to be relatively insensitive to the type of LiDAR scan, with the spatial and temporal filtering operated by the step-stare and continuous LiDAR being even

beneficial in some cases. In contrast, the error on the turbulence intensity exhibits a more consistent and opposite trend, with

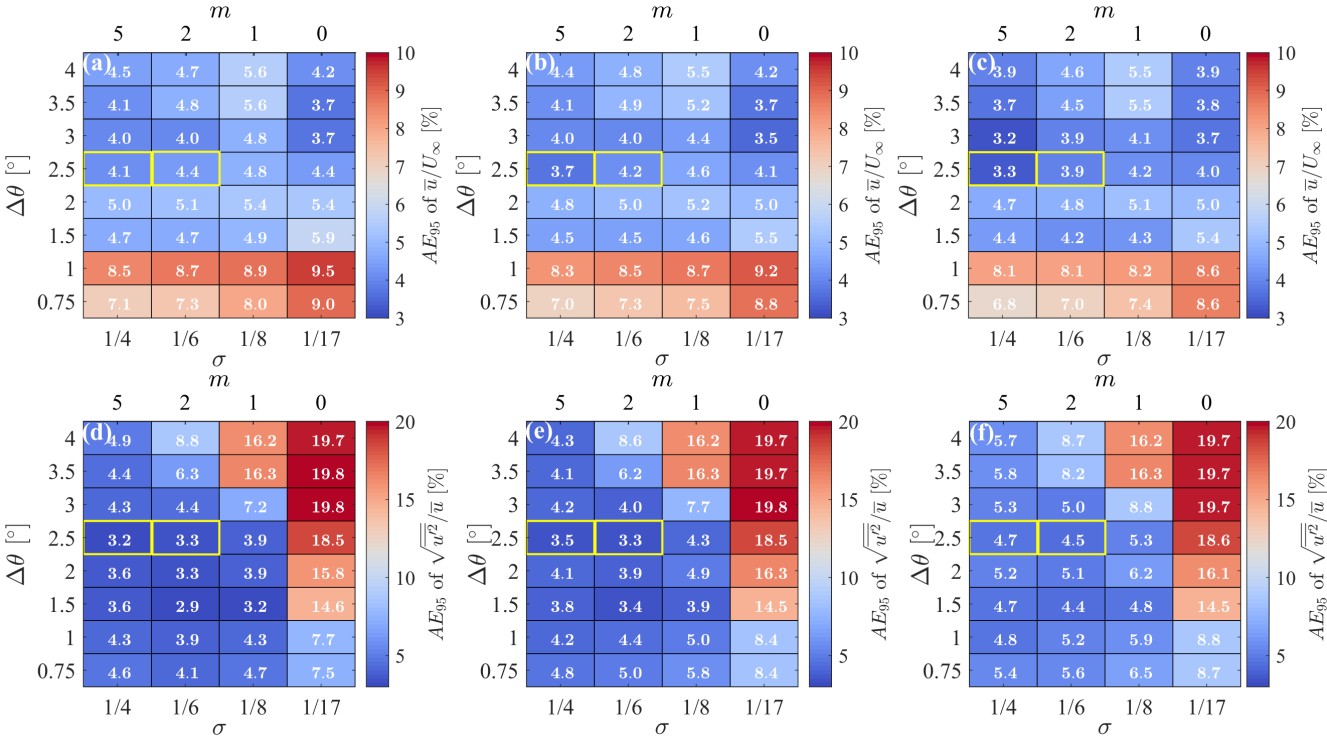

**Figure 5.** Error analysis of the LiSBOA applied to virtual radial velocity fields: **(a)** and **(d)** ideal LiDAR; **(b)** and **(e)** step-stare LiDAR; **(c)** and **(f)** continuous LiDAR; **(a)**, **(b)** and **(c)** mean streamwise velocity; **(d)**, **(e)** and **(f)** streamwise turbulence intensity. The optimal configurations are highlighted in yellow.





the continuous LiDAR showing the most severe turbulence damping. This feature has been extensively documented in previous studies, see e.g. Sathe et al. (2011).

This error analysis confirms that the optimal configurations selected through the Pareto front (i.e. $\Delta\theta = 2.5°$, $\sigma = 1/4 - m = 5$ and $\sigma = 1/6 - m = 2$) are arguably optimal in terms of accuracy ($AE_{95}$ of $\overline{u}/U_\infty = 3.3 - 4.1\%$ and $3.9 - 4.4\%$, $AE_{95}$ of

245 $\sqrt{\overline{u'^2}}/\overline{u} = 3.2 - 4.7\%$ and $3.3 - 4.5\%$, respectively) and data loss ($\epsilon^I = 33\%$ and $37\%$, respectively).

The 3D fields of mean velocity and turbulence intensity for first optimal configuration, (i.e. $\Delta\theta = 2.5°$, $\sigma = 1/4$, $m = 5$), are rendered in Figs. 6 and 7, respectively. Furthermore, in Fig. 8, azimuthally-averaged profiles at three downstream locations are also provided for a more insightful comparison. The mean velocity field is reconstructed fairly well regardless of the type of LiDAR scan, due to the careful choice of the fundamental half-wavelength, $\Delta n_0$, for this specific flow. On the other hand, the

250 reconstructed turbulence intensity is highly affected by the LiDAR processing, which leads to visible damping of the velocity variance for the step-stare and even more for the continuous mode. The ideal LiDAR scan, whose acquisition is inherently devoid of any space-time averaging, allows retrieving the correct level of turbulence intensity for locations for $x \geq 4D$, while in the near wake it struggles to recover the thin turbulent ring observed in the wake shear layer. Indeed, such short-wavelength feature has a small response for the chosen settings of the LiSBOA, in particular $\Delta n_0$ and $\sigma$ (see Fig. 3(b) of the companion

paper Letizia et al.). On the other hand, any attempt to increase the response of the higher-order moments, for instance by reducing the fundamental half-wavelengths or decreasing the smoothing and the number of iterations, would result in higher data loss and less experimental points per grid node.

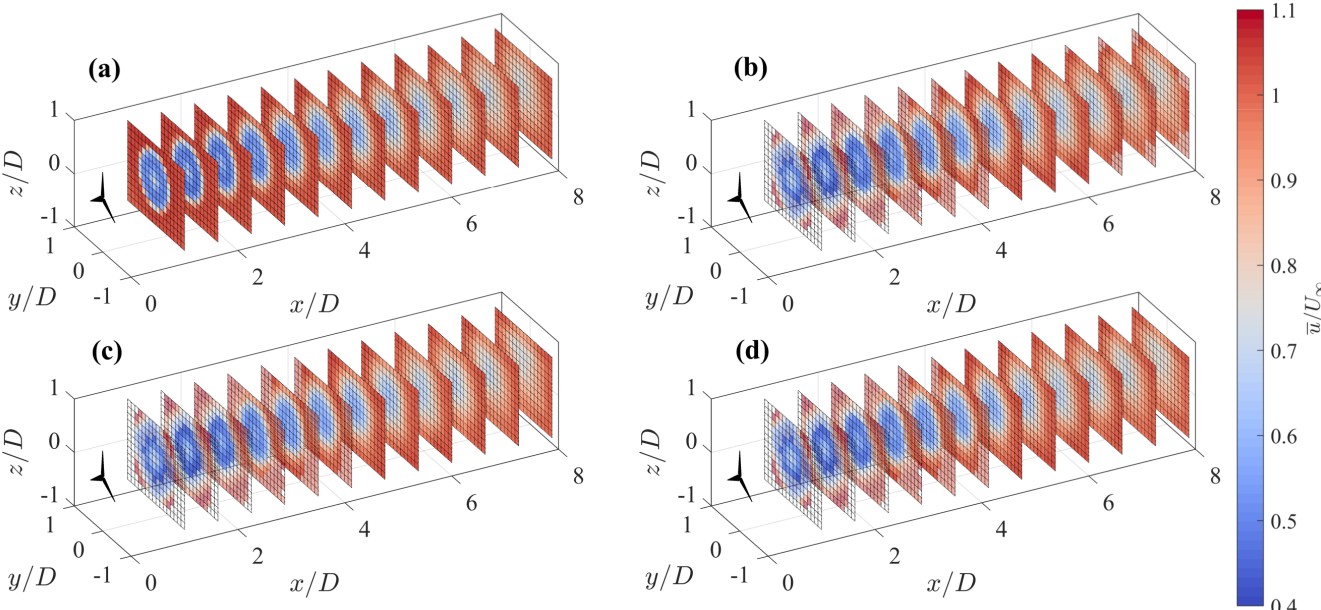

**Figure 6.** Mean streamwise velocity for $\Delta\theta = 2.5°$, $\sigma = 1/4$, $m = 5$: **(a)** LES; **(b)** ideal; **(c)** step-stare; **(d)** continuous mode LiDAR. The shaded area corresponds to the points rejected after the application of the Petersen-Middleton constraint.





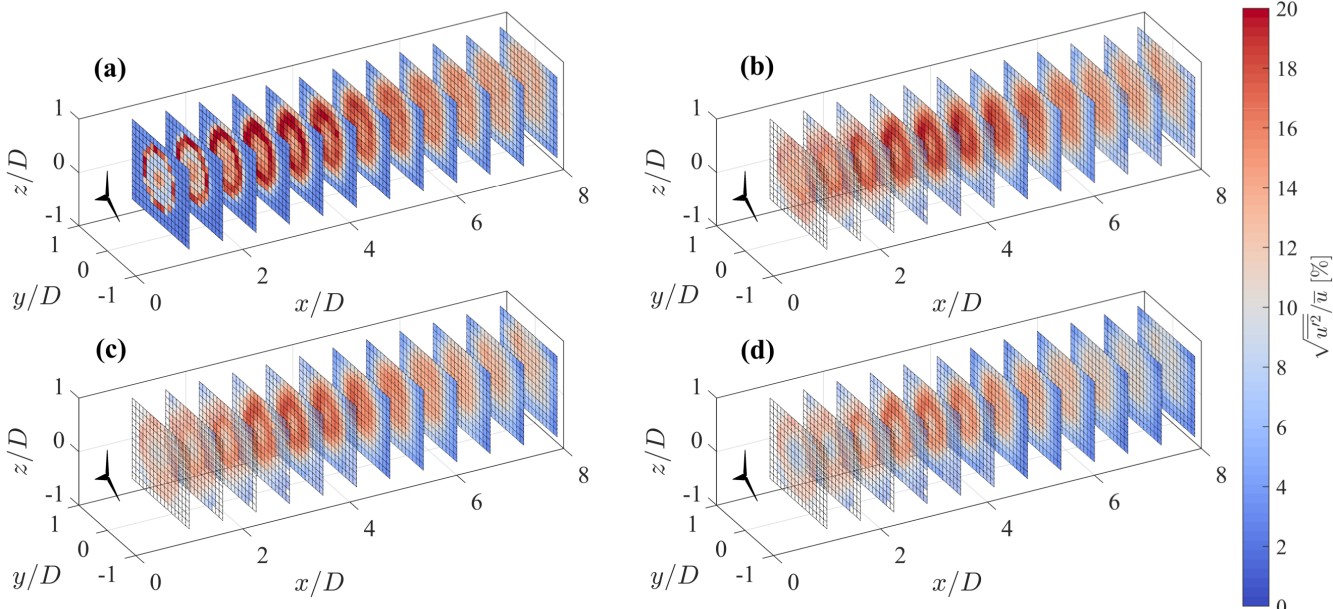

**Figure 7.** As in Fig. 6 but for streamwise turbulence intensity.

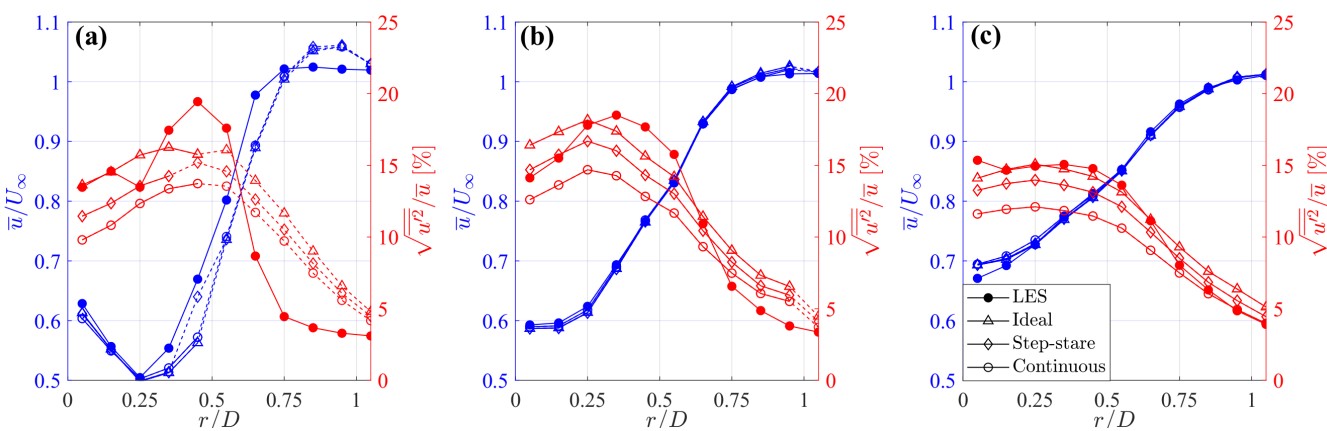

**Figure 8.** Azimuthally averaged profiles of mean streamwise velocity and turbulence intensity for three downstream locations: **(a)** $x/D = 2.25$; **(b)** $x/D = 4.125$; **(c)** $x/D = 6$. The dashed lines correspond to regions rejected after the application of the Petersen-Middleton constraint.

Finally, Figs. 9 and 10 show $\overline{u}/U_\infty$ and $\sqrt{\overline{u'^2}}/\overline{u}$ over several cross-flow planes and for all the combinations of $\sigma - m$ tested for the ideal LiDAR and the optimal angular resolution. For the mean velocity, the most noticeable effect is the increasingly 260 severe data loss consequent to the reduction of $\sigma$, which indicates $\sigma = 1/4$ - $m = 5$ as the most effective setting. The turbulence intensity exhibits, in addition to the data loss, a moderate increase in the maximum value for smaller $\sigma$, which is due to the





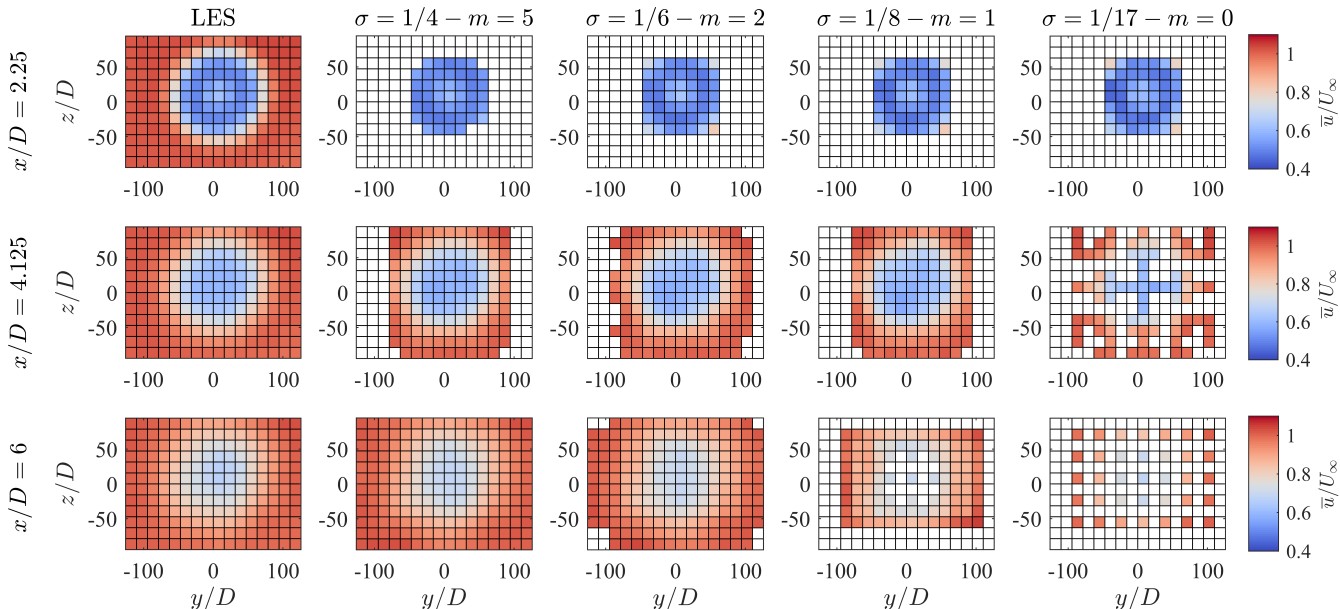

**Figure 9.** Mean streamwise velocity fields obtained through the ideal LiDAR simulator with $\Delta\theta = 2.5°$ over cross-flow planes at three downstream locations and four combinations of $\sigma - m$, compared with the corresponding LES data.

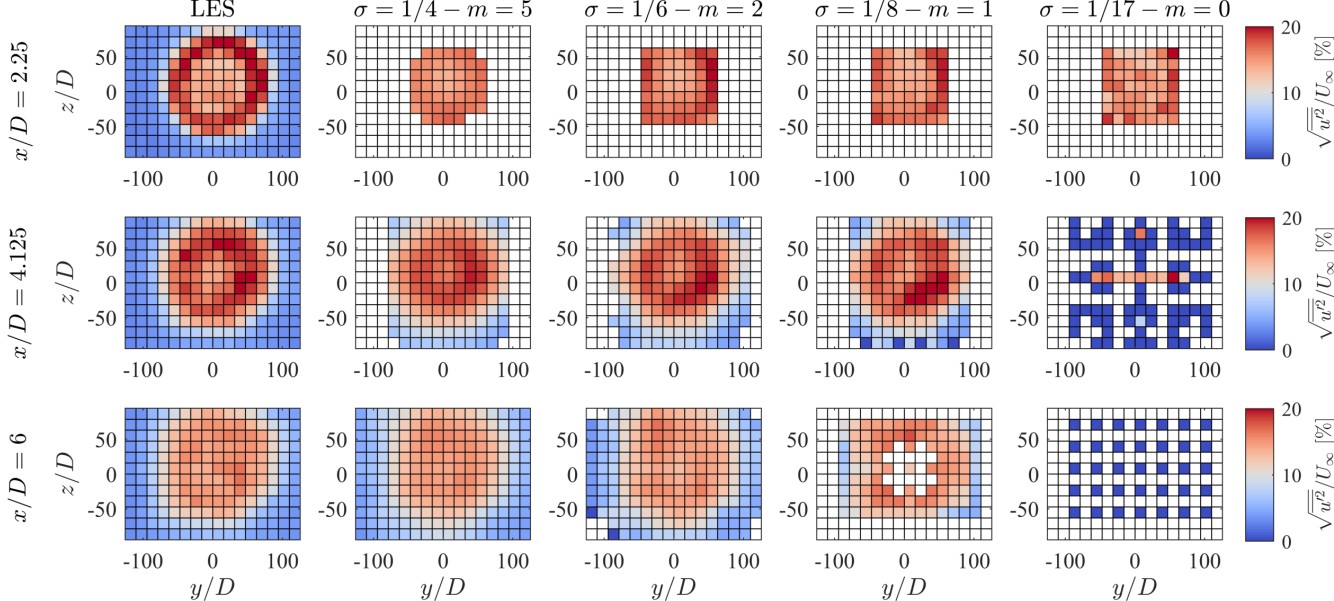

**Figure 10.** Same as Fig. 9 but for streamwise turbulence intensity.





higher response of the higher-order statistics (see Table 2 in Letizia et al.). However, this effect is negligible in the far wake were the radial diffusion of the initially sharp turbulent shear layer results in a shift of the energy content towards scales with larger $\Delta n$, that are fairly recovered even for $\sigma = 1/4$.

## 265   3   Application of the LiSBOA to wind LiDAR measurements

### 3.1   Site description and experimental setup

LiDAR data collected during an experimental campaign carried out at an onshore wind farm are used to assess the potential of the LiSBOA algorithm for wind energy applications. The measurements were collected during a long-term experimental campaign conducted at a large wind farm located in North-East Colorado (Fig. 11). This wind park encompasses 221 Mitsubishi 270   1-MW and 53 General Electric 1.5-MW wind turbines. More technical specifications of the wind turbines are provided in Table 1.

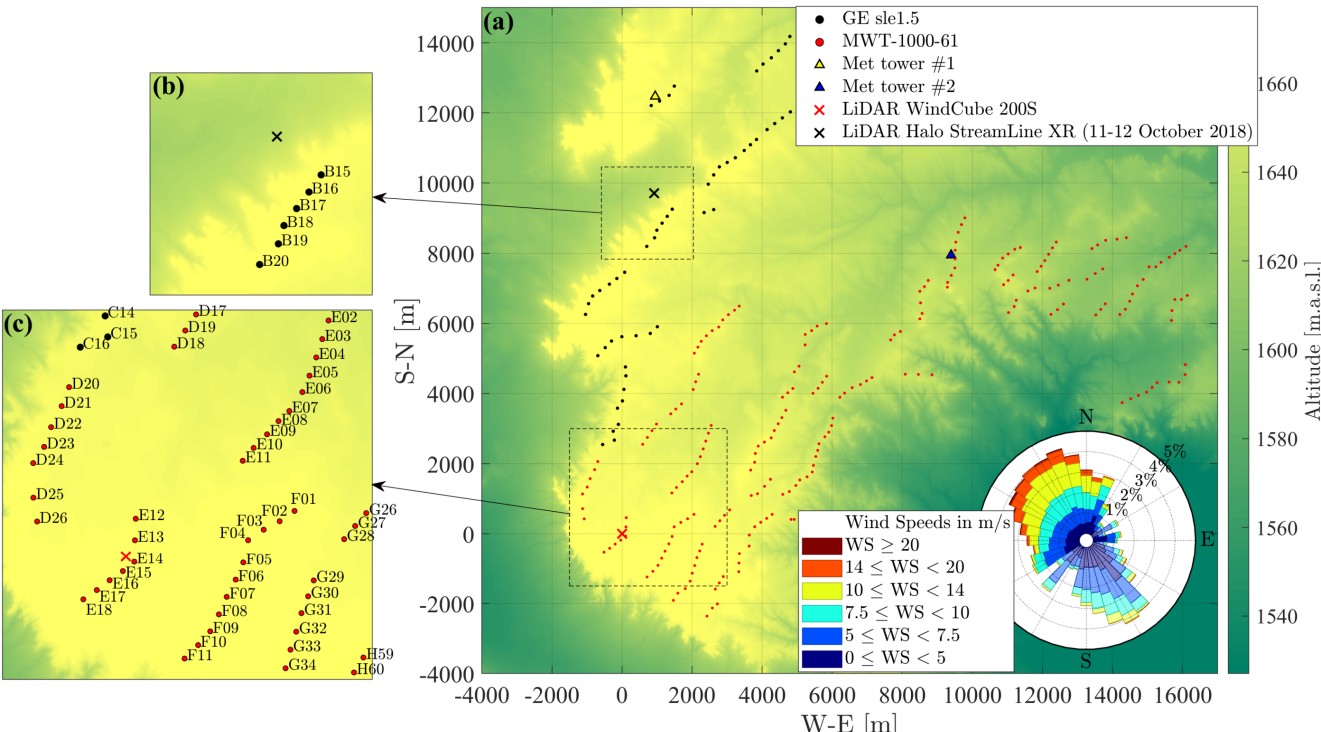

**Figure 11.** Map of the wind farm under investigation: **(a)** top view of the wind farm, with the diameter of the dots representing the turbine rotor diameter (in the wind rose, the sectors where both met-towers are potentially affected by turbine wakes are displayed in lighter color); **(b)** area probed through Streamline XR LiDAR on 11 and 12 October 2018; **(c)** typical field of view of the Windcube 200S LiDAR.





**Table 1.** Technical specifications of the wind turbines under investigation.

|  | MWT-1000-61 | GE sle1.5 |
| --- | --- | --- |
| Rated power [kW] | 1000 | 1500 |
| Cut-in wind speed [m s$^{-1}$] | 3.5 | 3.5 |
| Cut-out wind speed [m s$^{-1}$] | 25 | 25 |
| Rated wind speed [m s$^{-1}$] | 13.5 | 14 |
| Type | Variable pitch/fixed speed | Variable pitch/variable speed |
| Hub height [m] | 69 | 80 |
| Rotor diameter [m] | 61.4 | 77 |

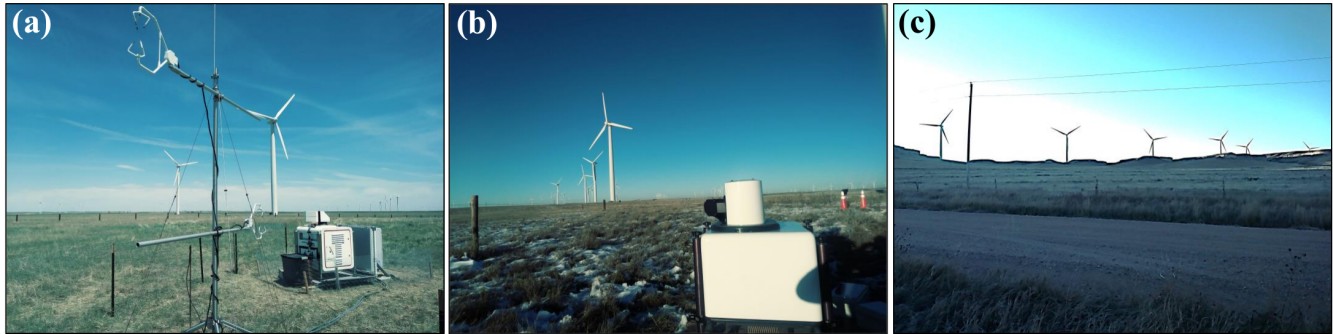

**Figure 12.** Photos of the LiDAR experiment: **(a)** LiDAR Windcube 200S and sonic anemometers Campbell Scientific CSAT3; **(b)** LiDAR Streamline XR; **(c)** GE 1.5sle turbines of the B row.

The wind rose, based on 3 years of wind speed and direction measured by the two meteorological (met) towers present on the site, reveals a prevalence of north-westerly and south-easterly wind directions. A characteristic of this site is the presence of a steep escarpment with an average jump in altitude of about 80 m surrounding a relatively flat plateau where the turbines are installed.

Two pulsed Doppler scanning wind LiDARs were deployed: a Windcube 200S manufactured by Leosphere (Fig. 12a) was installed for the period May-December 2018 in the southern part of the farm with the scope of detecting turbine wakes and flow distortions induced by the topography. The LiDAR was connected to the UTD mobile LiDAR station (El-Asha et al., 2017; Zhan et al., 2019) for remote control, scan setup, and data acquisition. Furthermore, a StreamLine XR by Halo-Photonics (Fig. 12b) was deployed for the period 11-19 October 2018 at specific sectors to investigate wake interactions and topography-related flow features. Additional details about the LiDARs are provided in Table 2.

The atmospheric stability is characterized through the Obukhov length (Monin and Obukhov, 1959) retrieved by two CSAT3 three-dimensional sonic anemometers manufactured by Campbell Scientific, which were deployed in the proximity of the UTD mobile LiDAR station at 1.4 m and 2.8 m above the ground. Two met-towers are installed in the northern part of the park, as



**Table 2.** Technical specifications of the wind LiDARs deployed during the field campaign.

|  | WindCube 200S | StreamLine XR |
| --- | --- | --- |
| Type | Pulsed - scanning | Pulsed - scanning |
| Scanning mode | Continuous | Step-stare or continuous |
| Wavelength [nm] | 1543 | 1500 |
| Pulse length [ns] | 200 | 200 |
| Frequency [kHz] | 10-40 | 10 |
| Minimum gate length [m] | 25 | 18 |
| Maximum range [m] | 6000 | 10000 |
| Maximum rotation speed [° s$^{-1}$ ] | 8 | 5 |
| Detection range [m s$^{-1}$] | $\pm$ 30 | $\pm$ 20 |

shown in Fig. 11. Each tower is equipped with 4 anemometers installed in paired configuration at heights of 50 m and 80 m for met tower #1, and 50 m and 69 m for met tower #2. Mean and standard deviation of wind speed and direction are stored every 10 minutes, along with the mean temperature and barometric pressure. In the present work, wind velocity data at each height are corrected for the flow distortion due to the tower following the guidelines provided by the IEC standards (International Electrotechnical Commission 61400-12-1 (2017) Annex G). Additionally, mean and standard deviation over

10-minute periods of nacelle wind speed, power, RPM, and blade pitch, collected and stored by the supervisory control and data acquisition (SCADA) system were made available. Normalized average power, $P_{\text{norm}}$, and $C_p$ curves based on the nacelle anemometers are built by leveraging data for the period 2016-2018, and shown in Fig. 13 as a function of the density-corrected normalized wind speed (International Electrotechnical Commission 61400-12-1, 2017):

$$U_{\text{norm}} = \frac{U_{\text{SCADA}}}{U_{\text{rated}}} \cdot \left( \frac{\rho_{\text{met}}}{\rho_{\text{ref}}} \right)^{1/3}, \tag{4}$$

where $\rho_{\text{ref}} = 1.225$ Kg m$^{-3}$ is the reference density at the sea level, $U_{\text{SCADA}}$ is the 10-minute average of the wind speed measured by the nacelle-mounted anemometers, while the local air density $\rho_{\text{met}}$ is calculated from the meteorological data according to the international standard (International Electrotechnical Commission 61400-12-1, 2017). Another important parameter derived form the SCADA data is the turbulence intensity at the rotor, defined as:

$$TI_{\text{SCADA}} = \frac{U_{\text{SD, SCADA}}}{U_{\text{SCADA}}} \tag{5}$$

where $U_{\text{SD,SCADA}}$ is the standard deviation of wind speed over 10-minute periods.

The two LiDARs performed a great variety of scans during the campaign, based on the specific phenomena under investigation. For the present analysis, we focus on the 3D reconstruction of non-interacting wakes using the high-resolution data collected with the Halo Streamline XR LiDAR and the 2D reconstruction of multiple overlapping wakes detected by the Windcube 200S.





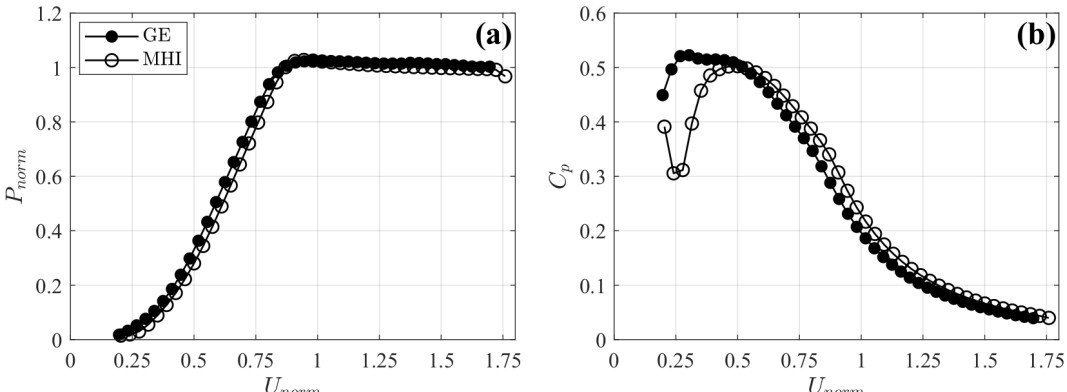

**Figure 13.** Performance curves for the General Electric and Mitsubishi wind turbines: **(a)** normalized power; **(b)** power coefficient, $Cp$.

## 3.2 Application of the LiSBOA to volumetric LiDAR data

The present section aims to explore the potential of the LiSBOA for the optimal design of a LiDAR experiment, data post-processing, and reconstruction of 3D flow statistics. The dataset used in the this section was collected on 11 October 2018 over the farm region shown in Fig. 11b through a StreamLine XR LiDAR. The goal of the experiment is to investigate the evolution of multiple turbine wakes advected over complex terrain. Figure 14 shows the site of the deployment and the relative distances between the LiDAR and the turbine hubs.

The deployment location was chosen to scan the wakes generated by the wind turbines B16-B19 for south-south-east wind directions. The LiDAR was deployed off a county road that connects the plateau with the surrounding plains, with a consequent difference in altitude between the instrument and the base of the turbines of about 40 m. To probe the wake region of turbines B16-B19 (Fig. 12b) and the leeward side of the ridge, seven PPI scans were performed by sweeping an azimuthal range of 65° with elevations angles, $\beta$, set to 5°, 6°, 7°, 8°, 10°, 12° and 15°. The total sampling time was selected equal to $T = 1h$, since the local weather forecast service provided by the wind farm operator predicted one hour of steady wind conditions blowing with SSE mean direction and speed of $U_\infty \approx 6$ m s$^{-1}$. The aerosol concentration allowed for the selection of a gate length of $\Delta r = 18$ m and accumulation time of 1.2 s.

As reported in Sect. 4 of Letizia et al., several parameters of the flow under investigation are required for the optimal design of the LiDAR scans. The fundamental half-wavelengths typical for wind turbine wakes were selected equal to those used in Sect. 2, i.e. $\Delta n_{0,x} = 2.5D$, $\Delta n_{0,y} = \Delta n_{0,z} = 0.5D$. Similarly, the integral time-scale was chosen equal to $\tau U_\infty/D$= 0.4 ($\tau \sim 5$ s). Finally, a measurement volume with dimensions of 1,000 m, 950 m, 130 m in the streamwise, transverse, and vertical directions, respectively, was selected to probe wakes generated from the turbines B16-B19 and the downwind region of the escarpment. The expected velocity standard deviation was estimated to be $\sqrt{\overline{u'^2}} = 0.125\, U_\infty$ based on previous field measurements of turbine wakes under stable conditions (Zhan et al., 2019).





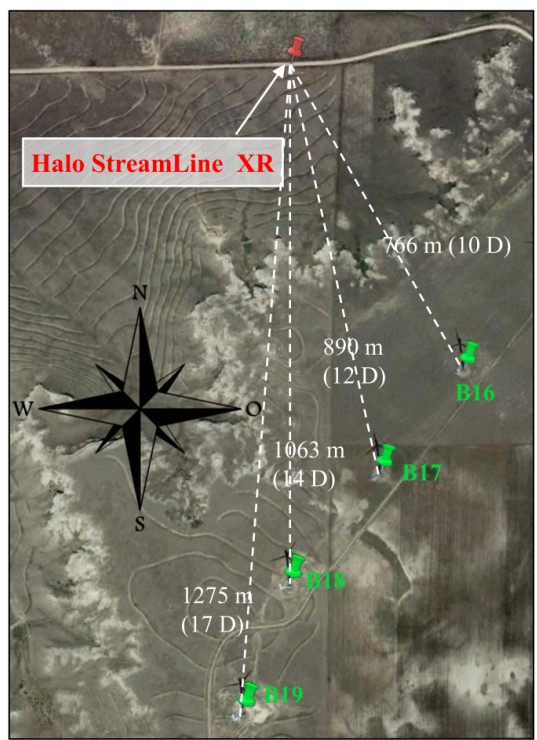

**Figure 14.** Satellite map of deployment of Halo StreamLine XR on October 11 2018. Source: Google Maps.

For the selection of the optimal azimuthal angular resolution of the LiDAR scan, the LiSBOA is applied to produce a Pareto front for six possible angular resolutions, $\Delta\theta$, between 0.25° and 4°, and four values of the smoothing parameter, $\sigma = [1/4, 1/6, 1/8, 1/17]$. As shown in Fig. 15, the optimal LiDAR scan is that with angular resolution $\Delta\theta = 1°$ and $\sigma = 1/4$ or $\sigma = 1/6$. Generally, an increasing $\Delta\theta$ entails a reduction of the standard deviation of the mean, $\epsilon^{II}$, yet values higher than $\Delta\theta = 1°$ do not lead to significant reductions of $\epsilon^{II}$ while worsening the data loss, $\epsilon^{I}$, indicating a larger number of grid points not satisfying the Petersen-Middleton constraint.

In Fig. 15, the values of the cost function $\epsilon^{I}$ and $\epsilon^{II}$ calculated from the LiDAR data after the quality-control process (Beck and Kühn, 2017) are also reported for the optimal angular spacing of the LiDAR $\Delta\theta = 1°$. It is noteworthy that there is negligible difference between the values calculated before and after the quality control of the LiDAR data, indicating that the data loss due to the acquisition error is negligible in the domain of interest. The spatial distributions of the grid points satisfying the Petersen-Middleton constraint for different values of $\Delta\theta$ and $\sigma = 1/4$ are reported in Fig. 16. It can be observed as $\Delta\theta = 1°$ represents the highest angular step ensuring an acceptable coverage of the spatial domain.

The data collected adopting the optimal scanning strategy with $\Delta\theta = 1°$ are now post-processed to calculate mean streamwise velocity and turbulence intensity. The time series of wind speed and direction recorded by the 4 anemometers installed on the met tower #1 and located at a distance of 2,700 m in the north direction from the test site are averaged to characterize



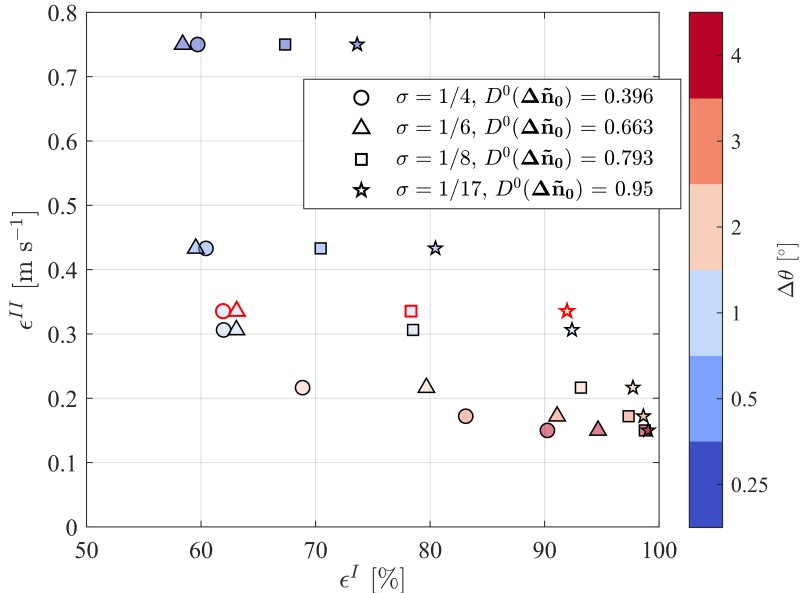

**Figure 15.** Pareto front for the design of the optimal LiDAR scan for the reconstruction of the wakes generated by the wind turbines B16-B19. The markers highlighted in red correspond to the respective parameters obtained from the actual LiDAR data after the quality control process.

the incoming wind. The evolution of wind speed and direction along with the velocity field measured with three specific PPI scans are reported in Fig. 17. For the time period between 20:30 and 21:30 local time (MDT) and indicated by the shaded area in Figs. 17a and b, the wind speed remained within the range between $5.1 \text{ m s}^{-1}$ and $7.1 \text{ m s}^{-1}$, while the wind direction departed less than $10°$ from its mean value of $\overline{\theta}_w = 163.4°$. The wind and power data, which are recorded by the SCADA

(Fig. 18), confirms that the turbines experienced fairly homogeneous inflow conditions, with differences in power capture 5% smaller than the rated value. The values of normalized velocity together with the performance curves (Fig. 13) indicate that the turbines were operating in region II of the power curve for the whole interval of interest.

Since statistical stationarity is an important assumption for the LiSBOA applications, adequate post-processing of the LiDAR data is needed to avoid effects on the reconstructed flow statistics due to the wind variability. Specifically, the wind speed

variability is corrected by making the line-of-sight velocity non-dimensional with the wind speed measured from the met tower #1. Furthermore, scans performed when the wind direction was outside of the range $\overline{\theta_w} \pm \Delta\theta_w$, with $\Delta\theta_w = 10°$, are excluded. After the quality control based on the dynamic filtering (Berg et al., 2011), 169,000 data points out of 455,000 are made available for the LiSBOA reconstruction on a Cartesian grid with resolution equal to $\boldsymbol{dx} = 0.25\boldsymbol{\Delta n_0}$. Isolated grid regions violating the Middleton-Petersen constraint ($< 2\%$ of the total number of grid points) are rejected and their respective values

are interpolated through Laplacian interpolation (*inpaint_nans.m* in Matlab). This analysis is restricted to the streamwise component of the wind velocity, which is estimated using the equivalent velocity approach (Zhan et al., 2019). The non-





**Figure 16.** Random data spacing, $\Delta\tilde{d}$, for 6 volumetric scans with different angular resolution and $\sigma = 1/4$. Points violating the Petersen-Middleton constraint ($\Delta\tilde{d} > 1$) are not displayed.

dimensional equivalent velocity is referred to as $\overline{u}/U_\infty$ in the remainder of the paper, while the associated turbulence intensity is referred to as $\sqrt{\overline{u'^2}}/\overline{u}$.

Figures 19 and 20 show 3D renderings of the non-dimensional velocity and turbulence intensity fields obtained by using the parameters $\sigma = 1/4$ - $m = 5$. Wake features, such as turbulent diffusion, the high-momentum jet in the hub region and the turbulent shear layer at the wake boundary, are well-captured. Two highly turbulent regions are located on both sides of the wakes, which is a distinctive signature of wake meandering occurring mostly horizontally in the ABL (España et al., 2011). The lack of symmetry and similarity among different turbines, however, suggests that full statistical convergence is not achieved

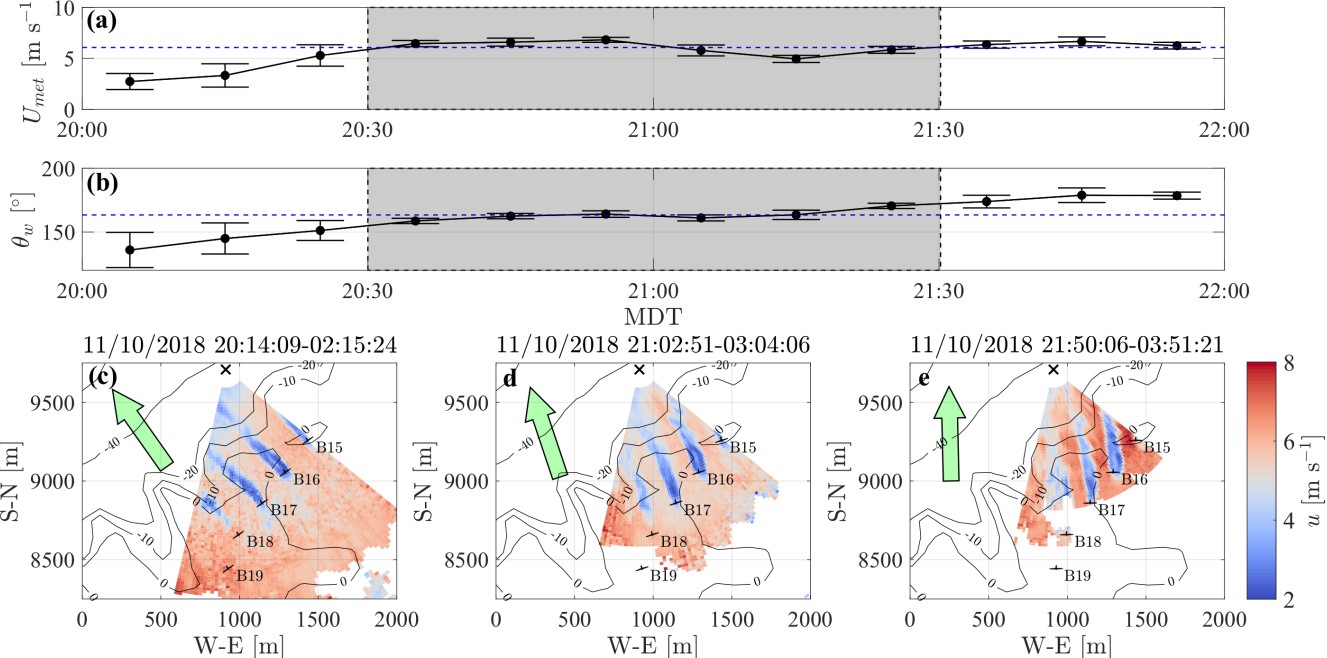

**Figure 17.** 3D LiDAR scans of five wind turbines: **(a)** 10-minute average wind speed measured from the anemometers installed at 50-m and 80-m height on the met-tower #1; the error bar represents the standard deviation over 10 minutes; the shaded area represents the interval selected for the LiSBOA application; **(b)** 10-minute average wind direction in geophysical reference system measured from the vanes installed at 50 m and 80 m on met-tower #1; **(c)**, **(d)** and **(e)** equivalent velocity fields measured with PPI scans; the green arrow is oriented as the mean wind direction measured by the met-tower #1, while the black cross indicates the LiDAR location.

on the second-order statistics for the available dataset. The low-speed region hovering over the down-slope represents most
probably the upper part of the low momentum zone that occurs past sharp escarpments (Berg et al., 2011).

The effect of the combination $\sigma - m$ on higher-order statistics is investigated by extracting the turbulence intensity at different cross-stream planes. The optimal pairs $\sigma - m$ identified by the Pareto front analysis (Fig. 15), viz. $\sigma = 1/4$ - $m = 5$ and $\sigma = 1/6$ - $m = 2$, are tested here. One may expect that due to the difference in the response of the high-order moments of the fundamental mode between the two pairs, $D^0(\mathbf{\Delta \tilde{n}_0})$, the first case would exhibit a significantly lower $\sqrt{\overline{u'^2}}/\overline{u}$ with respect to second one.
However, as shown in Fig. 21, the peaks of turbulence intensity are quite similar between the two cases. The main difference between the two reconstruction processes is a smoother distribution of $\sqrt{\overline{u'^2}}/\overline{u}$ for $\sigma = 1/4 - m = 5$. The similarity between the two cases is due essentially to two reasons: first, the smallest energy-containing length scales of the turbulence intensity field (i.e. shear layer thickness) are larger than the selected fundamental mode $\Delta n_{0,y} = \Delta n_{0,z} = 0.5D$; second, the larger number of points per grid node averaged for the $\sigma = 1/4$ case, leads to a higher variance due to the reduction of the bias of the
estimator of the variance, which partially compensates the lower theoretical response. Summarizing, this sensitivity analysis suggests that the choice of the $\sigma - m$ pair cannot be based purely on the theoretical response, since it does not take into account





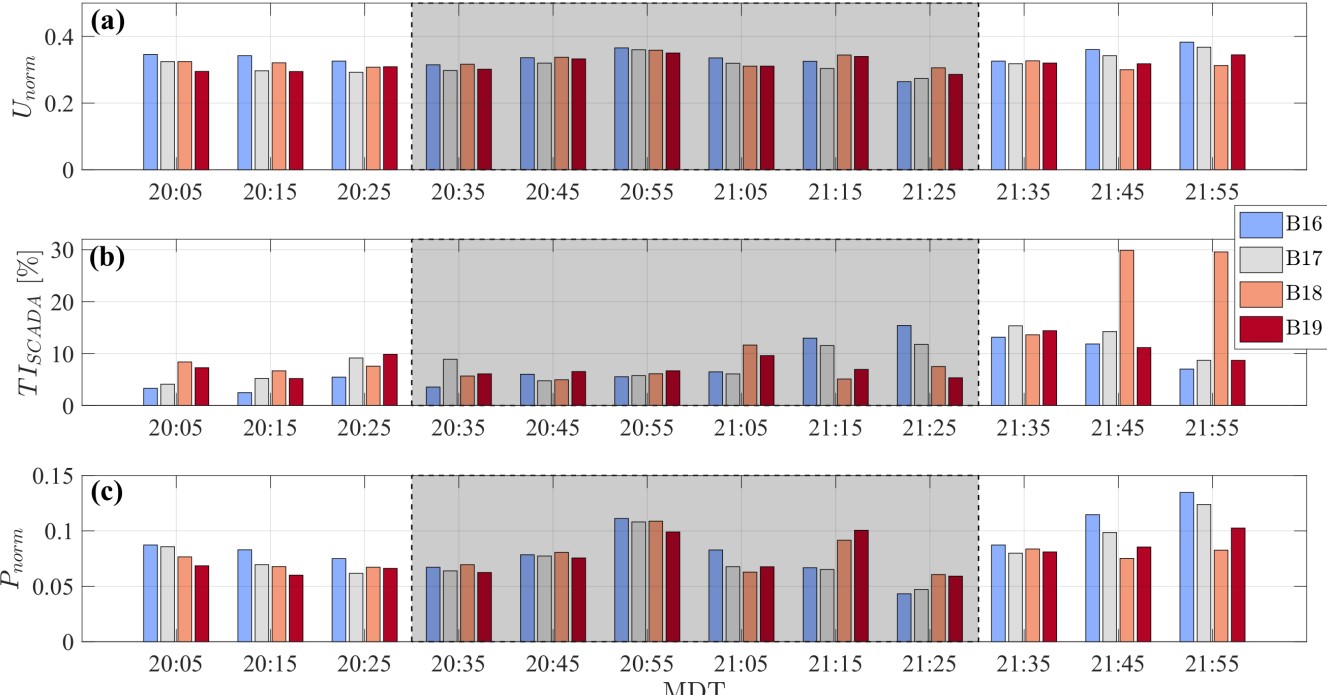

**Figure 18.** SCADA data during the selected testing period: **(a)** normalized hub-height velocity; **(b)** turbulence intensity; **(c)** normalized power.

non-ideal effects deriving for the discrete and non-uniform data distribution. Instead, an a posteriori analysis of the statistics retrieved is recommended to select the best $\sigma - m$ values.

Turbine-wake statistics are extremely sensitive to the width of the selected wind sector (Barthelmie et al., 2009; Hansen
and Barthelmie, 2014). It is well-known that widening the wind direction range can lead to an enhanced wake diffusion and turbulence intensity (Trujillo et al., 2011; Kumer et al., 2015), compensated by higher data availability and statistical significance. A sensitivity analysis to the wind sector width for reconstructing the statistics through the LiSBOA for two additional values of $\Delta\theta_w$ is now presented. Besides the baseline value of $10°$, effects of a narrower ($\Delta\theta_w = 5°$) and wider ($\Delta\theta_w = 15°$) range are investigated. The standard deviation of the wind direction associated with the different sectors is $1.08°$,
$1.93°$ and $2.74°$ for $\Delta\theta_w = 5°, 10°, 15°$, respectively. Figure 22 show the rotor-averaged velocity and turbulence intensity for each turbine as a function of the downstream distance from the rotor. The profiles of the mean and standard deviation obtained for different $\Delta\theta_w$ are practically the same, indicating that the effects of wind direction variability on wake flow statistics are not significant.

For the sake of completeness, the velocity and turbulence intensity sampled in the cross-stream plane where the maximum
velocity deficit occurs ($x/D \sim 1.3$) for all the turbines and the $\Delta\theta_w$ are shown in Fig. 23. The discrepancies due to different $\Delta\theta_w$ are negligible. A more evident mismatch can be observed in the shape of the wakes among different wind turbines, with

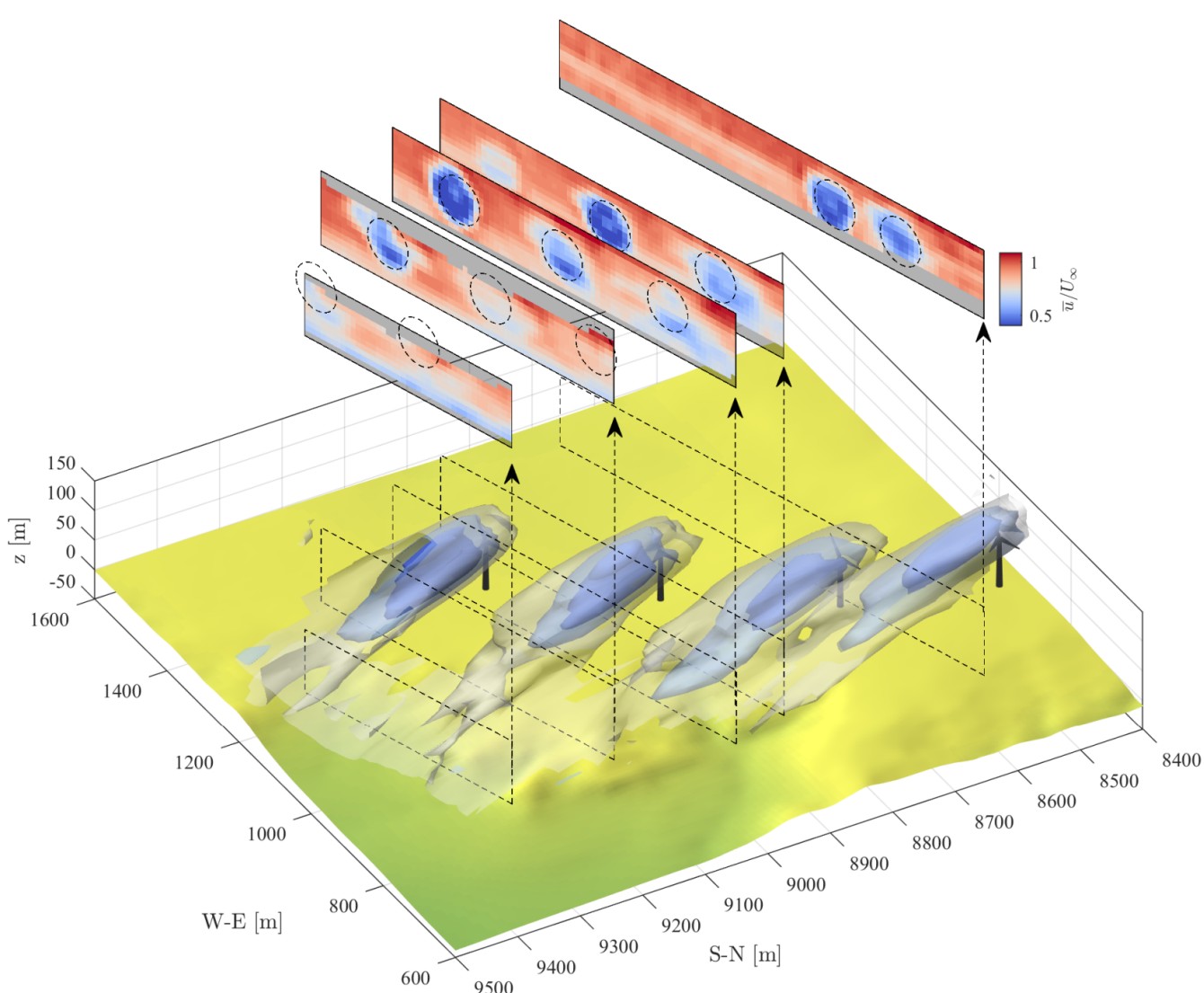

**Figure 19.** 3D rendering of the normalized mean equivalent velocity field reconstructed with $\Delta\theta_w = 10°$. The three isosurfaces represent $\overline{u}/U_\infty = 0.45$, 0.6 and 0.75, while the color maps represent cross-sections of the mean velocity field over the respective planes reported in the rendering. The dashed circles correspond to the rotor swept area of turbines B16-B19 (from left to right) projected onto the specific cross-plane.

the wake of turbine B19, in particular, showing the velocity deficit and turbulence peak that are displaced above the hub height. Turbine B19 is also the only one facing a slightly inclined terrain (see Fig. 22), which may have caused a skewed inflow.

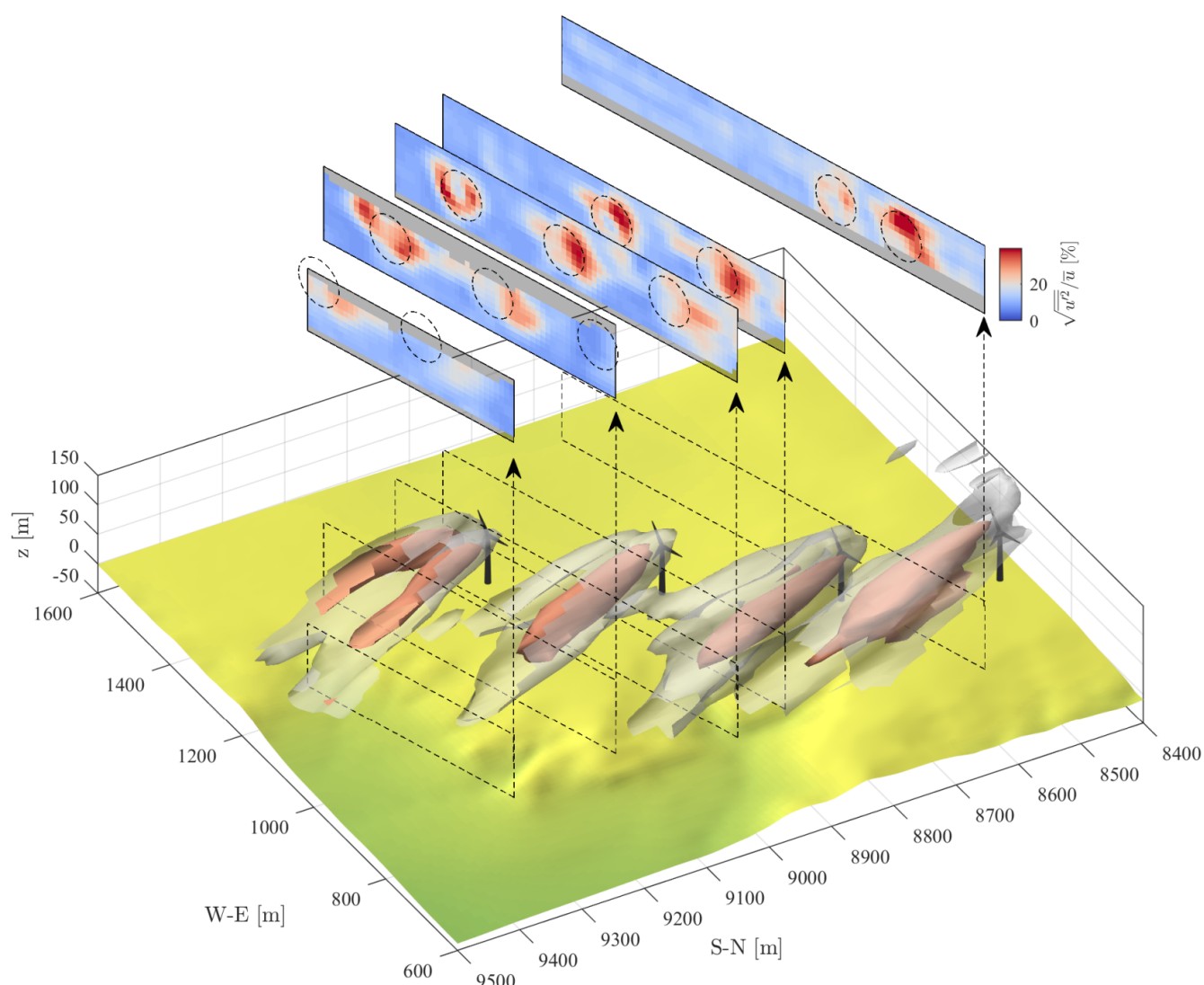

**Figure 20.** 3D rendering of the turbulence intensity field reconstructed with $\Delta\theta_w = 10°$. The two isosurfaces represent $\sqrt{\overline{u'^2}}/\overline{u} =$ levels of 20% and 30%, while the color maps represent cross-sections of the turbulence intensity field over the respective planes reported in the rendering. The dashed circles correspond to the rotor swept area of turbines B16-B19 (from left to right) projected onto the specific cross-plane.

## 3.3 Application of the LiSBOA to interacting wind turbine wakes

An assessment of the accuracy of the LiSBOA in the calculation of mean wind speed and turbulence intensity is now provided for LiDAR measurements performed during the occurrence of wake interactions. To this aim, point-wise measurements provided by the nacelle-mounted anemometers and saved in the SCADA data of four closely spaced Mitsubishi wind turbines,



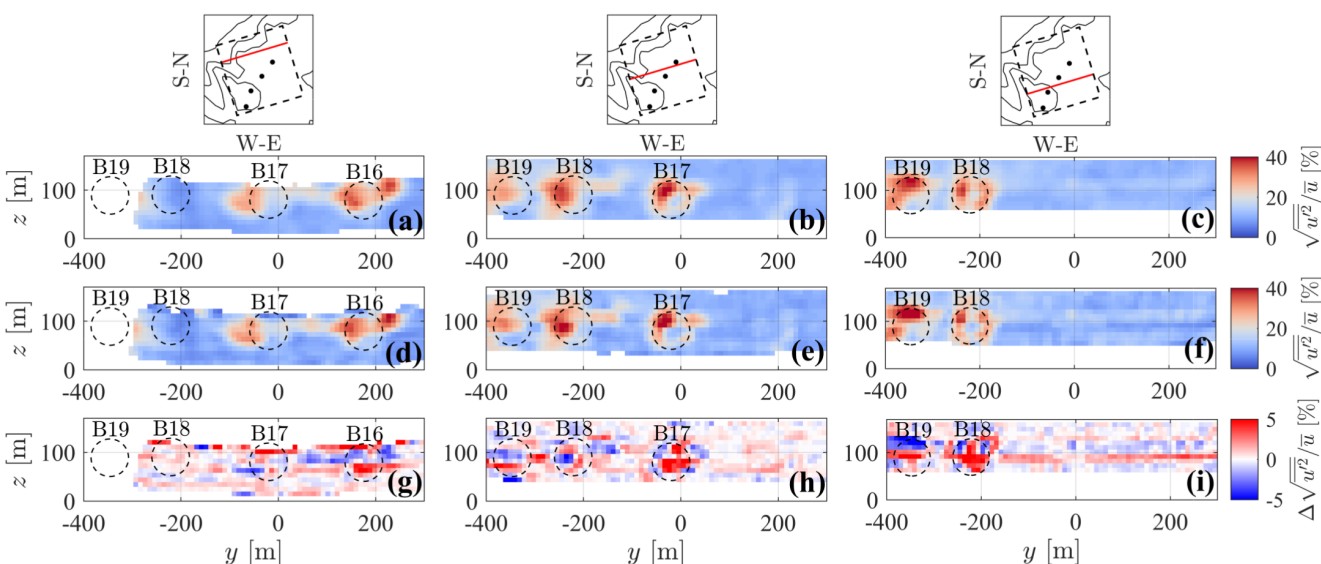

**Figure 21.** Comparison of the turbulence intensity reconstructed with $\sigma = 1/4 - m = 5$ **(a, b, c)** vs. $\sigma = 1/6 - m = 2$ **(d, e, f)** and their difference **(g, h, i)** for three selected streamwise locations indicated by the red lines in the top maps.

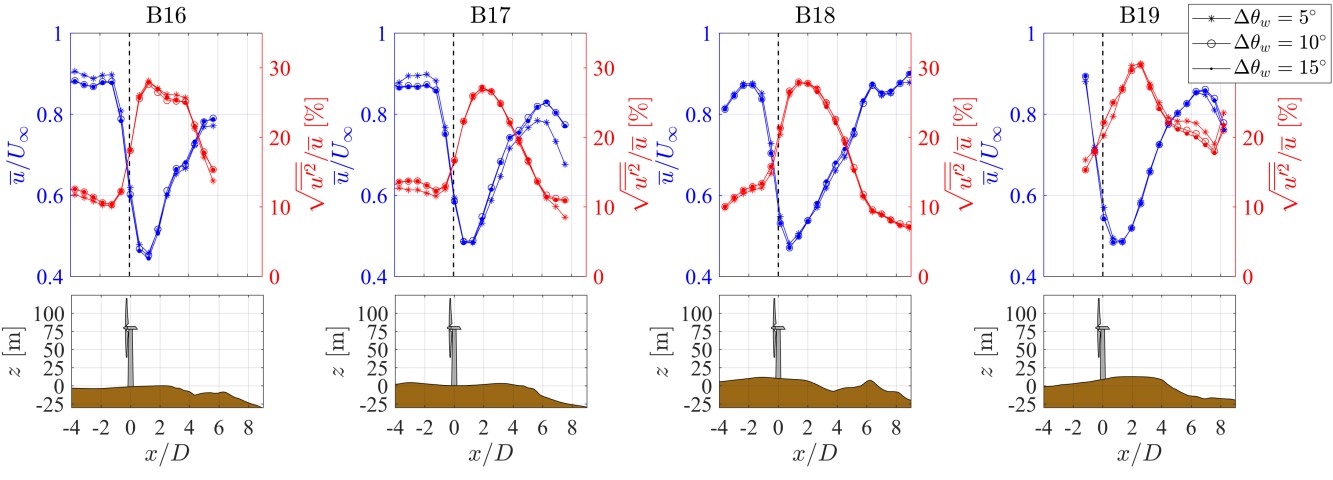

**Figure 22.** Rotor-averaged streamwise mean velocity and turbulence intensity as a function of the downstream distance from the turbine and associated altitude profile.

roughly aligned with the wind direction, are compared with the statistics obtained from the post-processing of the LiDAR data with the LiSBOA.

Figure 24 reports a satellite image of the site of this experiment. The tests were performed during the occurrence of a nearly steady north-easterly wind ($U_\infty \sim 8$ m s$^{-1}$) from 9 pm to 1 am local time ($T = 4h$) in the night between 5 and 6 September





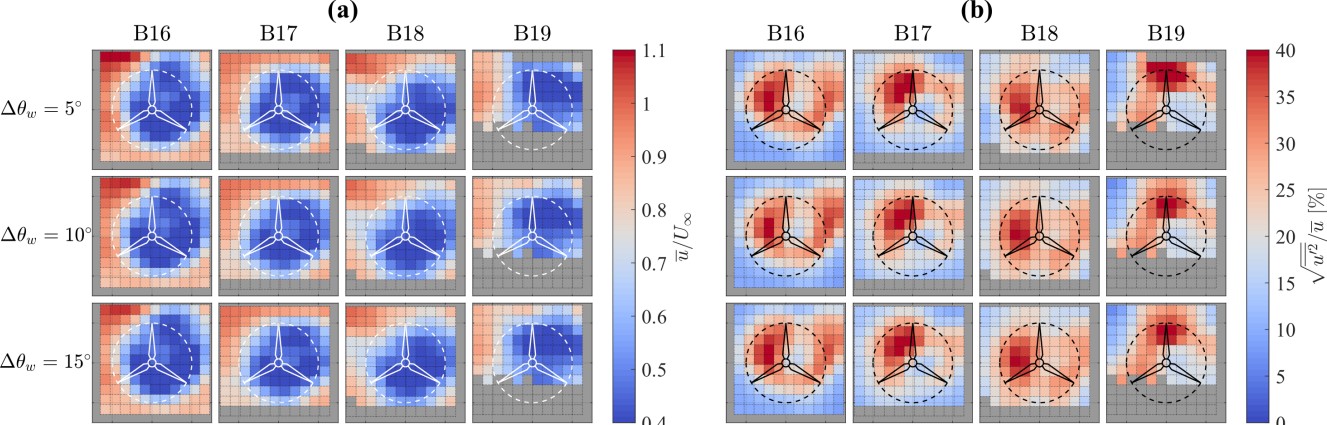

**Figure 23.** Fields reconstructed adopting several $\Delta\theta_w$ values and sampled at $x/D \sim 1.3$ downstream of turbines B16, B17, B18 and B19: **(a)** mean streamwise velocity; **(b)** streamwise turbulence intensity.

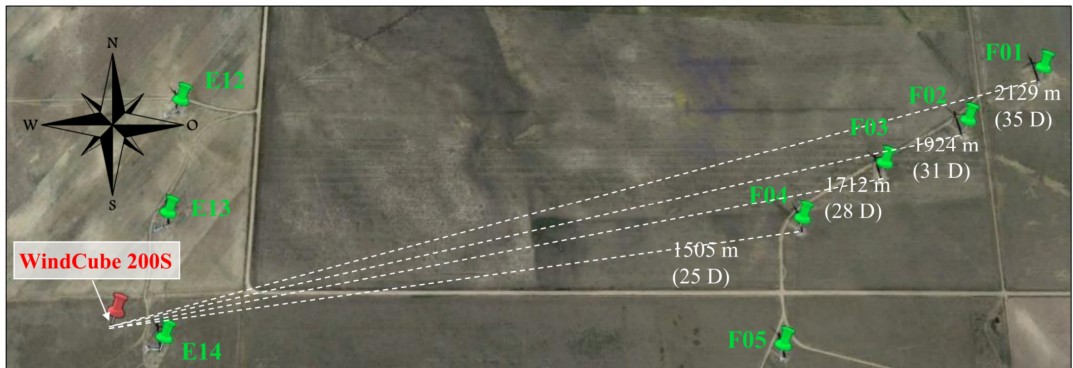

**Figure 24.** Satellite map of the site for the deployment of the Windcube 200S LiDAR, including the four Mitsubishi wind turbines under investigation. Source: © Google Maps.

2018. This wind condition created a good alignment of the wakes emitted by the turbines F01 to F04. The aerosol conditions allowed us to run the Windcube 200S LiDAR with a gate length of 50 m and an accumulation time of 0.5 s. The LiDAR is located at a distance of about $25D$ from the wind turbine F04, which is the most downstream turbine for that specific wind

condition, while the average streamwise spacing between the turbines is $3.6D$. The velocity and turbulence intensity fields are reconstructed over a horizontal plane including only points within the vertical range spanning from the bottom- to top-tip of the turbine rotors. The 2D reconstruction here adopted implies that a uniform weight is applied for points displaced at different $z$, which means the reconstructed statistics represent time and vertically-averaged fields. This 2D approach is deemed convenient for the comparison with point-wise measurements recorded by the SCADA through nacelle-mounted instruments representing

an average of the wind characteristics over the rotor.



The region of interest was probed through a volumetric scan consisting of three PPI scans with elevation angles $\beta = 2.1°, 2.6°$, and $3.3°$. The fundamental half-wavelengths were selected as $\Delta n_{0,x} = 2.25D$, $\Delta n_{0,y} = 0.75D$. Accordingly to the previous cases, the integral time-scale was estimated to be $\tau U_\infty / D = 0.4$ ($\tau \sim 3$ s). The velocity variance was set to $\sqrt{\overline{u'^2}} = 0.2\, U_\infty$. The value of the associated turbulence intensity is higher than that used for non-overlapping wakes to ac-

count for the turbulence build-up, which is known to occur for turbines operating experiencing wake interactions (Chamorro and Porté-Agel, 2011; Iungo et al., 2013a).

The incoming wind is characterized by averaging measurements collected from all the anemometers and wind vanes installed on both met towers, which are located 12 km and 10.4 km away from the leading turbine F01 (Fig. 25). The Obukhov length is calculated from both sonic anemometers indicating a stable stratification regime. The SCADA data exhibits the typical

signature of multiple wake interactions with reduced wind speed and power for downstream turbines, while turbulence intensity is enhanced, in particular for the F02 and F04 wind turbines.

The optimal design of the LiDAR scan is performed considering six values of $\Delta\theta$ and four values of $\sigma$. The obtained Pareto front is shown in Fig. 26, which indicates $\Delta\theta = 0.5°$ and $\sigma = 1/3, 1/4$ or $1/6$ as the optimal scanning parameters. The equivalent velocity retrieved by the LiDAR is made non-dimensional with the freestream velocity provided by the met-towers.

The wind direction range is set to $\Delta\theta_w = 10°$, resulting in a total measuring period of 150 minutes. Data points lying above the top-tip or below the bottom-tip heights are excluded for this data analysis. The dynamic filter technique is used to reject corrupted LiDAR data, producing a total of 544,000 quality-controlled LiDAR samples over 1,327,000 collected LiDAR data within the selected wind-direction range.

The LiSBOA is carried out on a grid with resolution $\boldsymbol{dx} = 0.25\boldsymbol{\Delta n_0}$, using the combination smoothing parameters - number

of iterations $\sigma = 1/6$ - $m = 1$, which is, among the allowable combinations, the one providing the largest response of the higher-

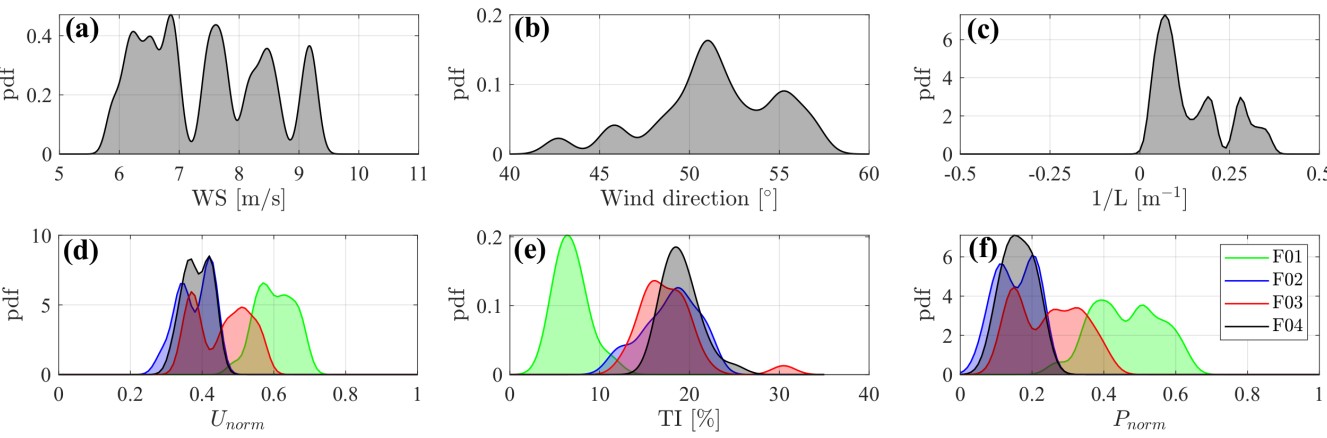

**Figure 25.** Probability density functions of the met and SCADA data recorded from 21:00 to 1:00 MDT on the night between September 5 and 6 2018: **(a)** wind speed from met-towers; **(b)** wind direction from met-towers; **(c)** inverse Obukhov length from our sonic anemometers; **(d)** normalized wind speed from SCADA; **(e)** turbulence intensity from SCADA; **(f)** normalized power from SCADA.





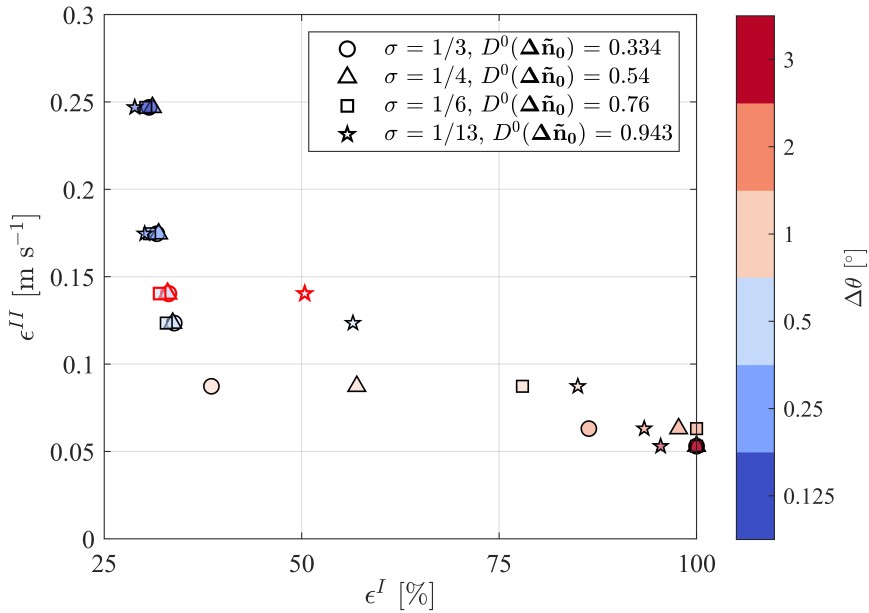

**Figure 26.** Pareto front for the design of the optimal LiDAR scan for the reconstruction of the wakes statistics for the turbine F01-F04. The markers highlighted in red represent the actual LiDAR data after the quality control.

order moments. The obtained velocity and turbulence intensity fields over the horizontal plane at hub height are displayed in Fig. 27. The velocity deficit of F02 appears slightly larger than that detected behind the unwaked turbine F01, which is most probably due to the wake superimposition. An even deeper velocity deficit can be observed behind F03, which operates in a partially waked condition for this specific wind direction. Downstream of the third turbine, the wake deficit build-up saturates,

confirming results from previous studies on close wake interactions (Barthelmie et al., 2010; Chamorro and Porté-Agel, 2011). Finally, the relatively fast recovery of the wake of the trailing turbine, F04, can be ascribed to the enhanced mixing due to the wake-generated turbulence. Indeed, Fig. 27b shows significant wake-generated turbulence increasing past the leading turbine that reaching its maximum at a distance of $1D$ downstream of the rotor of F03. Interestingly, wake-generated turbulence is concentrated on the sides of the wake of F01, which experiences undisturbed flow, while it spreads among the whole wake

region for the downstream turbines. This feature might be related to the presence of coherent wake vorticity structures in the near wake of turbine F01 (Iungo et al., 2013a; Viola et al., 2014; Ashton et al., 2016), while further downstream, the perturbed inflow promotes the breakdown of such coherent structures leading to more homogeneous turbulence. Finally, the large velocity deficit/high turbulence detected in the wake of F03 may be a consequence of the mentioned partial wake interaction, which exposes the rotor to a non-homogeneous flow resulting during off-design operations.

From a more quantitative standpoint, the incoming wind conditions experienced by each turbine are characterized to perform a direct comparison with the nacelle-anemometer data. To this aim, the mean velocity and turbulence intensity profiles are extracted from the LiDAR statistics at a distance of $1D$ upstream of the rotors over a segment spanning the whole rotor



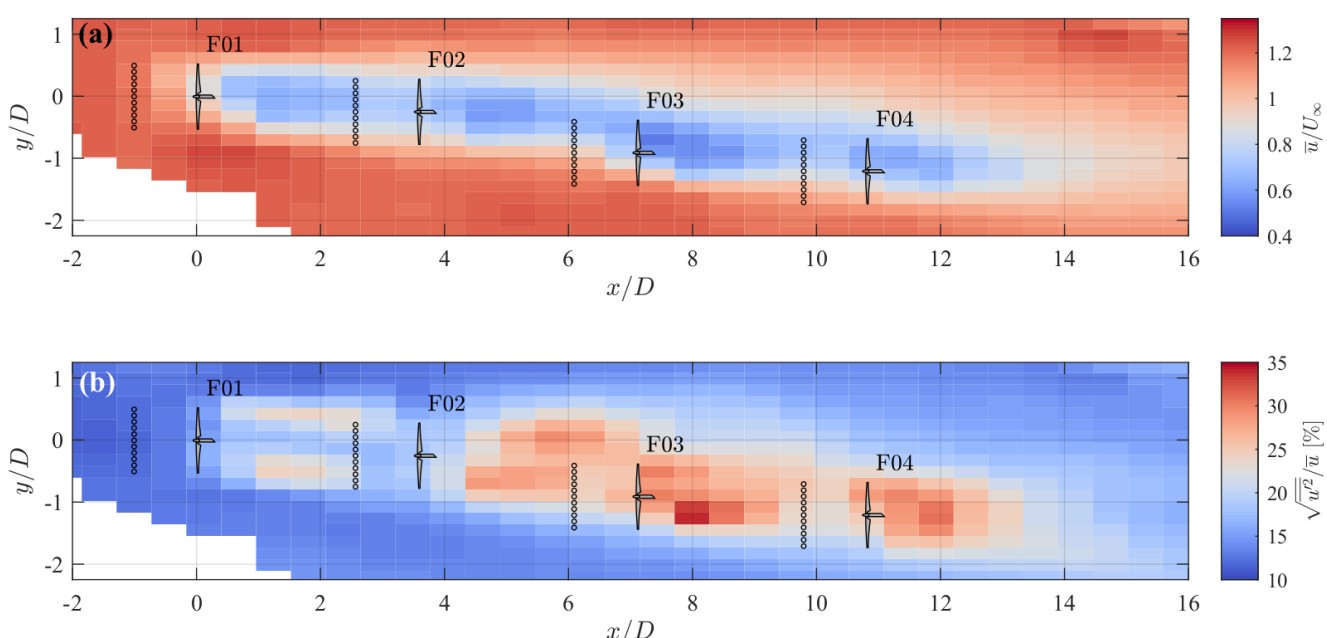

**Figure 27.** Velocity statistics of the wakes generated by the turbines F01-F04 reconstructed over the horizontal plane at hub height: **(a)** mean streamwise velocity; **(b)** streamwise turbulence intensity. The black dots indicate the sampling locations used for the estimation of the incoming flow for the respective turbine.

diameter. The sampling location is chosen based on previous studies (Politis et al., 2012; Hirth et al., 2015), since $1D$ is generally considered the minimum distance upstream of the rotor where the influence of the induction zone can be neglected

for normal operative conditions. The averaged values of $\overline{u}/U_\infty$ and $\sqrt{\overline{u'^2}}/\overline{u}$ of each upstream profile are then used for the comparison with the respective values recorded through the SCADA.

A well-posed comparison of the wind statistics obtained from the LiSBOA, the SCADA and met-data requires two important elements: firstly, the statistical moments compared have to be equivalent; secondly, both the LiSBOA and the SCADA data must be representative of the freestream conditions experienced by each turbine.

Regarding the first issue, the mean field obtained through the LiSBOA, $\overline{u}$, can be expressed as:

$$\left\langle \frac{u}{U_\infty} \right\rangle_T = \left\langle \left\langle \frac{u}{U_\infty} \right\rangle_{\hat{T}} \right\rangle_T \sim \left\langle \frac{U_{\text{SCADA}}}{U_{\text{met}}} \right\rangle_T \tag{6}$$

where $\langle . \rangle_T$ is the average calculated over the whole sampling period of 150 minutes, while $\langle . \rangle_{\hat{T}}$ is the 10-minute average performed by the SCADA and the met-tower acquisition system. $U_{\text{SCADA}}$ and $U_{\text{met}}$ are the 10-minute averaged velocities recorded from the SCADA and met-tower, respectively, while the symbol $\sim$ indicates statistical equivalence.



Similarly, for the comparison between the velocity variance calculated through the LiSBOA and the respective values recorded through the SCADA, we have the following relationship:

$$\left\langle \frac{u'^2}{U_\infty^2} \right\rangle_T = \left\langle \left\langle \frac{\hat{u}'^2}{U_\infty^2} \right\rangle_{\hat{T}} \right\rangle_T + \left\langle \left\langle \frac{u}{U_\infty} \right\rangle_{\hat{T}}^2 \right\rangle_T - \left\langle \left\langle \frac{u}{U_\infty} \right\rangle_{\hat{T}} \right\rangle_T^2 \sim \left\langle \frac{U_{\mathrm{SD,\,SCADA}}^2}{U_{\mathrm{met}}^2} \right\rangle_T + \left\langle \frac{U_{\mathrm{SCADA}}^2}{U_{\mathrm{met}}^2} \right\rangle_T - \left\langle \frac{U_{\mathrm{SCADA}}}{U_{\mathrm{met}}} \right\rangle_T^2 \qquad (7)$$

where $u'$ and $\hat{u}'$ are the velocity fluctuations with zero mean calculated over the time periods $T$ and $\hat{T}$, respectively. The parameter $U_{\mathrm{SD,\,SCADA}}^2$ is the velocity variance recorded by the SCADA over the period $\hat{T}$ of 10 minutes.

To ensure that the SCADA mean and standard deviation of velocity are representative of the undisturbed wind conditions at each rotor, these velocity statistics are corrected for the flow distortion induced by the turbine through appropriate nacelle transfer functions (NTF), which converts the velocity statistics measured at the nacelle of a wind turbine to the corresponding freestream values measured from a met-tower located nearby. The IEC standard 61400-12-2 (International Electrotechnical Commission, 61400-12-2, 2013) prescribes to calculate the NTF from the bin average with bin size 0.5 m s$^{-1}$ of the velocity measured by a reference anemometer as a function of the nacelle wind speed. In the present work, besides correcting the mean wind speed as indicated by the IEC standards, a linear correction of the wind speed standard deviation is also applied, as suggested by Argyle et al. (2018). We adopted as reference anemometer that installed at 69 m above the ground on met tower #2. The SCADA data of Mitsubishi turbines H05 and H06, both falling in the range of distances from the met-tower recommended by the IEC 61400-12-1 (International Electrotechnical Commission 61400-12-1, 2017), are used. Only the

unwaked wind sectors calculated based on the same standard are considered. The described layout is shown in Fig. 28, while Fig. 29 shows the result of this analysis. There is a high correlation between the velocity measured by the met-tower and the nacelle-mounted anemometer ($\rho = 0.976$). Nevertheless, the NTF of the velocity reveals consistently lower values occurring at the nacelle compared to the met-tower, with a peak at 20 m s$^{-1}$. Concerning the standard deviation of velocity, the agreement between SCADA and met-tower data is significantly lower ($\rho = 0.828$), yet a linear correction can be still calculated with

acceptable significance (error on slope and intercept are 0.0038 and 0.0034 with 95% confidence).

The results of the comparison between LiSBOA and SCADA are provided in Fig. 30. The mean velocity is accurately captured and confirms that F02 and F04 are the turbines mainly affected by the upstream wakes. The slightly higher momentum impinging F03 is mostly due to the imperfect alignment of that rotor with the upstream turbine wakes, which creates a condition of partial-wake interaction. A slightly larger discrepancy between LiSBOA and SCADA data is observed for the turbulence in-

tensity, with a maximum difference of $\sim 3\%$ for F03. Nonetheless, the main trend is well reproduced and the overall agreement is satisfactory. The observed difference in turbulence intensity can be related to several factors, such as turbulence damping due to the LiDAR measuring process and LiSBOA calculations, the accuracy of the NTF, estimate of the streamwise velocity from the LiDAR radial velocity or vertical dispersive stresses.

The effect of the sampling location upstream of the turbines in the LiSBOA field is investigated by quantifying the discrep-

ancy of the LiSBOA statistics with respect to the reference SCADA values for all the turbines through the 95-th percentile of the absolute error, $AE_{95}$. Figure 31 shows $AE_{95}$ as a function of the distance upstream where the incoming flow is extracted from the LiSBOA statistics. For the mean velocity, it is confirmed that the value suggested by the literature ($x = -1D$) is sufficiently far from the rotor to limit the effects of the induction zone on the definition of the reference freestream velocity.





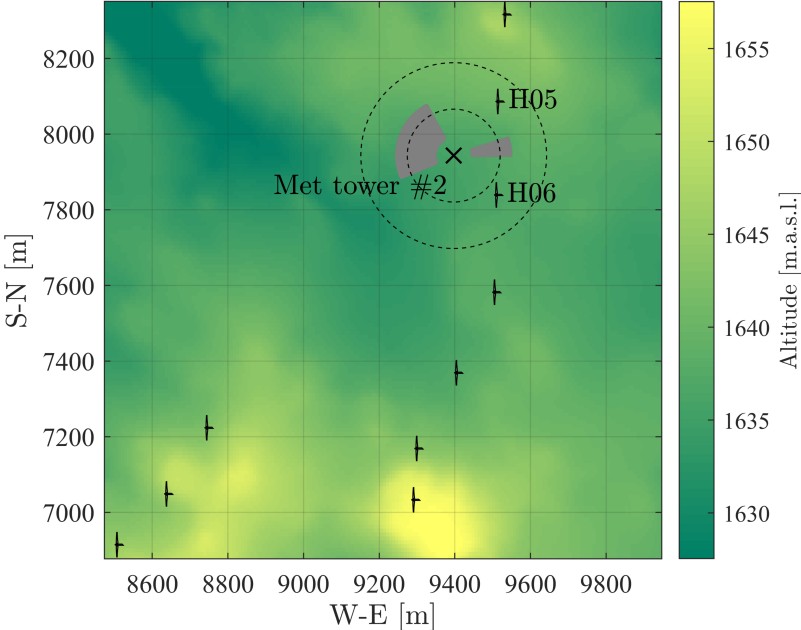

**Figure 28.** Met-tower and turbines selected for the nacelle transfer function estimation. The directions highlighted in grey represent the valid wind sectors unaffected by turbine wake interactions. The dashed circles bound the allowed range of distances from the tower in compliance with IEC standard 61400-12-1 (International Electrotechnical Commission 61400-12-1, 2017).

Furthermore, the rotor thrust does not seem to have noticeable effects on the incoming turbulence, being the induction zone
essentially devoid of significant turbulent fluctuations due to the loads of the turbine blades. The discrepancy between the turbulence intensity retrieved through LiSBOA and SCADA steeply increases for sampling locations further than $2D$ from the rotor.

Summarizing, the satisfactory agreement between LiSBOA and SCADA data achieved in the present study indicates the proposed procedure as a promising candidate for wind resource assessment, especially for complex terrains, and investigations
of the intra-wind-farm flow.

## 4    Conclusions

The LiDAR Statistical Barnes Objective Analysis (LiSBOA) has been applied to three different cases of wind turbine wakes to estimate the optimal LiDAR scanning strategy and retrieve mean velocity and turbulence intensity fields.

The first dataset objectively analyzed has been generated through the virtual LiDAR technique, namely by numerically
sampling the turbulent velocity field behind the rotor of a 5 MW turbine obtained from a large eddy simulation (LES). The 3D mean normalized streamwise velocity and turbulence intensity fields have shown a maximum error with respect to the LES statistics of about 4%. Wake features, such as the high-velocity stream in the hub region and the turbulent shear layer at





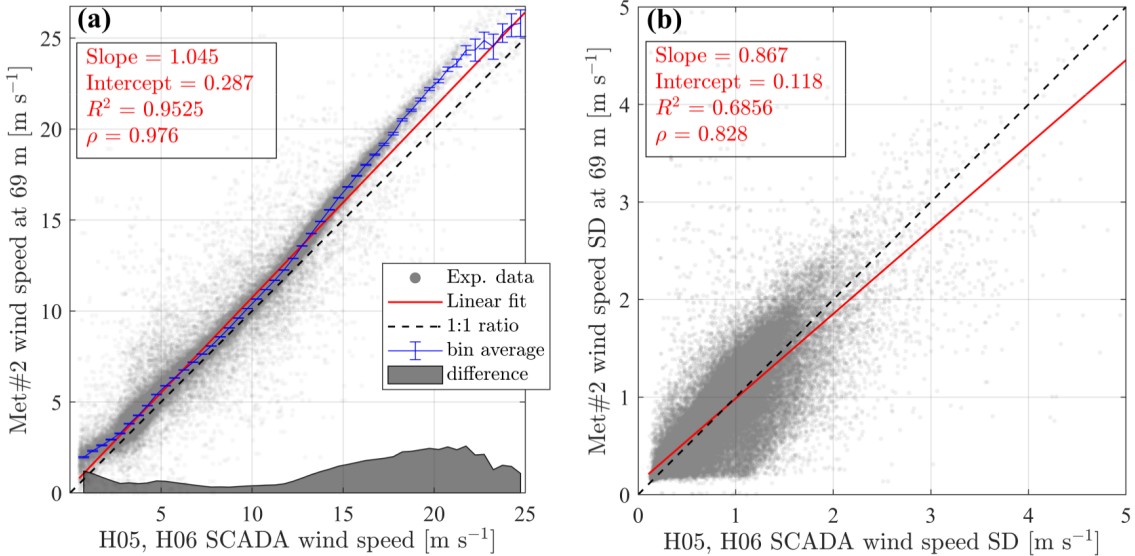

**Figure 29.** Nacelle transfer function for mean **(a)** and standard deviation **(b)** of wind speed. The error bars represent the standard error on the mean with 95% confidence level.

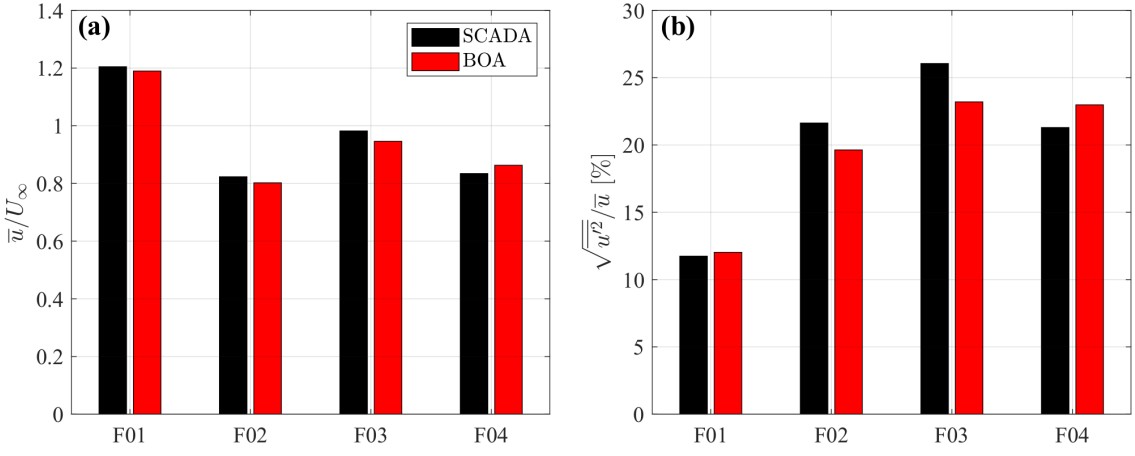

**Figure 30.** Comparison between LiSBOA and SCADA wind statistics for a case with wake interactions: **(a)** mean streamwise velocity normalized by freestream velocity; **(b)** streamwise turbulence intensity.

the wake boundary, have been accurately reconstructed. This analysis has also confirmed that the optimal scanning strategy identified by the LiSBOA has been that producing the most accurate flow statistics.

Subsequently, the LiSBOA has been used to process real LiDAR data collected for a utility-scale wind farm. For the first test case, the statistics of the wakes of four non-interacting 1.5-MW turbines placed at the brink of an escarpment have been reconstructed. The optimal LiDAR scanning strategy has been selected through the LiSBOA, while the mean velocity and





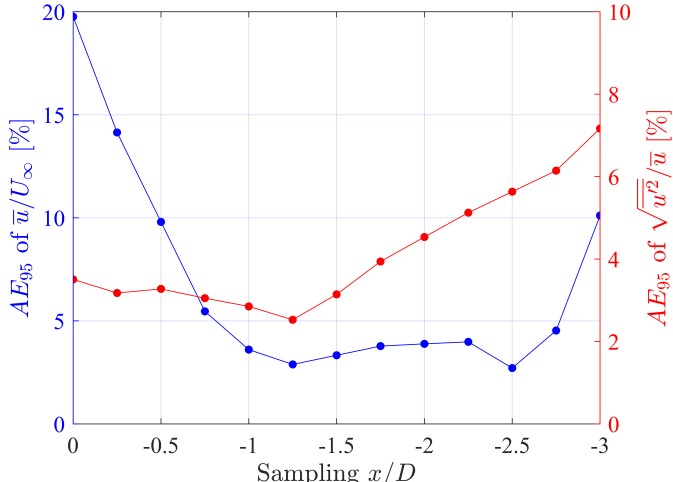

**Figure 31.** $AE_{95}$ of mean velocity and turbulence intensity for F01-F04 as a function of the upstream sampling location of the LiSBOA fields.

turbulence intensity fields retrieved through the LiSBOA have offered a detailed insight of the wake morphology. Furthermore, a sensitivity analysis of the wind direction range has confirmed the robustness of the data selection and quality control methods.

Finally, the complex velocity field arising from the interaction of four 1-MW turbines has been analyzed by calculating first and second-order moments on the horizontal plane. The mean velocity and turbulence intensity extracted $1D$ upstream of the rotors have agreed well with the values provided by the nacelle anemometers, with maximum discrepancies as low as $3\%$.

The applications of the LiSBOA discussed in this work aims to showcase the potential of the proposed procedure for the optimal design of LiDAR scans and to provide guidelines for the utilization of the LiSBOA for the analysis of LiDAR data.
Two noticeable advantages of the LiSBOA arise from the present work: first, the LiSBOA allows a straightforward yet effective design of LiDAR scans, which exploits only basic knowledge about the flow under investigation and the LiDARs used. This feature can be of interest, especially when planning field experiments that involve multiple LiDARs, complex topography, or articulated turbine configurations. In such situations, the use of the proposed quantitative and comprehensive scan design approach may be beneficial to narrow down a great deal of arbitrariness and uncertainty associated with the campaign planning.
Second, the LiSBOA offers complete control over the response of the spatial wavelengths of the velocity field for the statistical moments with various order. This feature is crucial when dealing with turbulent and multi-scale flows because it allows extracting meaningful information from the flow while filtering out small-scale variability.

*Code availability.* The LiSBOA algorithm is implemented in a publicly available code which can be downloaded at the following URL: https://www.utdallas.edu/windflux/.



*Author contributions.*   SL and GVI developed the LiSBOA and prepared the manuscript. The LiDAR data were generated as a team effort including contributions from all three authors. SL implemented the LiSBOA in a Matlab code and performed data analysis under the supervision of GVI.

*Competing interests.*   No competing interests are present.

*Acknowledgements.*   This research has been funded by a grant from the National Science Foundation CBET Fluid Dynamics, award number
1705837. This material is based upon work supported by the National Science Foundation under grant IIP-1362022 (Collaborative Research: I/UCRC for Wind Energy, Science, Technology, and Research) and from the WindSTAR I/UCRC Members: Aquanis, Inc., EDP Renewables, Bachmann Electronic Corp., GE Energy, Huntsman, Hexion, Leeward Asset Management, LLC, Pattern Energy, EPRI, LM Wind, Texas Wind Tower and TPI Composites. Any opinions, findings, and conclusions or recommendations expressed in this material are those of the authors and do not necessarily reflect the views of the sponsors. Texas Advanced Computing Center is acknowledged for providing
computational resources. The authors acknowledge the support of the owners and operator of the wind farm, and the land owners of the test site.





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
