# Peer review of "LiSBOA: LiDAR Statistical Barnes Objective Analysis for optimal design of LiDAR scans and retrieval of wind statistics. Part II: Applications to LiDAR measurements of wind turbine wakes"

_Atmospheric Measurement Techniques, 2020_

## Referee Comment (RC1) · Anonymous Referee #1 · 9 Oct 2020

In this study, the authors build on the theoretical work discussed and presented in the companion part 1 paper for reconstructing the wind fields downstream of wind turbines to measure the properties of the turbine wakes. Specifically, the velocity deficit and turbulence intensity are measured. The authors first demonstrate this capability using a virtual lidar simulator to quantify the expected errors, then also demonstrate the capability on measurements near a wind farm in Colorado. The results look compelling, and there is some comparison with in situ measurements to validate the wind field

reconstruction where anemometers were installed. Overall, this is a nice demonstration of the novel technique and the analysis of the wind turbine wakes will be of interest to those in the wind energy field. Still, the virtual lidar simulator needs to be revised as there are several modifications that could be made to it to obtain more realistic results, which will yield a more accurate understanding of how to interpret real-world measurements. This analysis will require significant additional data analysis. Thus, I recommend major revisions to this manuscript after which it may be acceptable for full publication in AMT.

Specific Comments

a) Line 15: It would be helpful to include all the symbols used in the paper in this list, not just those used in LiSBOA.

b) Line 127: This should be projection of the wind vector, not velocity, onto the laser beam to really represent a lidar measurement.

c) Eq. 1: What is u here? Since there is no arrow over it, I'll assume it is just the streamwise component of the wind within the LES simulator, and not the full 3-D vector. To truly simulate a measurement, it should be the full 3-D wind vector as the radial velocity is not only affected by the streamwise component, but also the vertical and crosswise components (whose means are zero, but instantaneous turbulent perturbations are not). This may have significant effects on the results.

d) Line 157: Why is a freeslip enforced on the bottom of the domain? That does not produce a realistic logarithmic wind profile.

e) Line 164-166: The text becomes very confusing to this reader around here. The authors should make it clear that the optimal design of the lidar scan is based on the flow characteristics. Thus, the flow characteristics shown and discussed in the next several paragraphs come from the raw LES field. It might be helpful to make the analysis of the LES flow statistics its own subsection to provide clear separation from

the lidar simulator itself. It was confusing to me to see lidar simulator results in Fig. 1 immediately followed by analysis of the LES field, before returning to the lidar simulator again. Sect. 2 could benefit from some reorganization as well, to mitigate alternating between the two separate subjects.

f) Line 177: Is the integral time scale calculated using a time series of the streamwise velocity in the LES field?

g) Line 185, Fig 2, Fig. 3: Clarify what is meant by the spectra (and other features) are averaged azimuthally. What does that mean exactly?

h) Line 203: Just to be clear, the constant angular resolution $\Delta\theta$ is for both azimuth and elevation, correct? That is $\Delta\theta = \Delta\beta$.

i) Line 225: State the equation for the equivalent velocity approach.

j) Fig. 6/7 (and discussion of it): It would be help to indicate over how much time these statistics are computed over. Based on the statistics, I think it's 160 sec but I may be wrong.

k) Sect 2: Doppler wind lidar measurements are subject to error that increases with decreasing SNR; as SNR typically decreases with range, the velocity measurement also becomes less accurate. This error should be considered within the wind lidar simulator for more realistic results of true measurements.

l) Line 350: Clarify how the wind speed variability is corrected by making the LOS velocity non-dimensional, this is not obvious to the reader.

m) Figure 17: The timestamps above each PPI plot (panels c-e) are confusing and should be removed. It's unclear why each time stamps spans >6 hours.

Editorial Corrections

a) Line 315: Need a space between 65deg and with.

---

## Referee Comment (RC2) · Anonymous Referee #2 · 13 Oct 2020

This manuscript applies the LiSBOA algorithm introduced in the companion paper (Part I) to retrieve wind speed and turbulence intensity from Doppler lidar measurements in wind turbine wakes. A LES data set is used to investigate the quality of LiSBOA algorithm and LiSBOA retrievals from ground-based Doppler lidar measurements are compared to anemometer measurements.

This study is within the scope of AMT, but there are some major issues that need to be

addressed before it can be accepted. Overall, this manuscript is rather long and has a large number of figures. I suggest to move the LES discussion (Section 2) to Part I (where it can replace the synthetic data) to keep this manuscript focused on the actual measurements.

Major comments

L160 Some measurements have been conducted with lidars located at the nacelle, but in this study only ground-based lidars are utilised. Please use the LES to assess the quality of ground-based lidar measurements.

L165-198 Determining the flow characteristics for LiSBOA is not trivial, as seen here and in Figs. 2-3. Please test LiSBOA sensitivity to the flow characteristics. In atmospheric applications they may not be known with good enough accuracy, or they may vary during measurements.

Specific comments

L17 "angular resolution" which angle?

L156 Please state resolution in metres.

L157 What is "A radiative condition is imposed at the outlet"?

L157 "freeslip is enforced on the top and bottom" This doesn't create a realistic wind profile. It may not hamper the use of the LES to validate LiSBOA algorithm but should be justified in the text.

Fig. 1. Please state the integration time per radial measurement for panels b-d.

L211 Please make it clear that the aliasing here is an issue for LiSBOA retrieval, not for the lidar itself.

L215 What is the total sampling time?

L226 Please define "equivalent velocity approach".

L227 "turbulence intensity" could be defined at the first use.

Table 2. Please give the actual operating specifications used in this study. E.g. "Scanning mode Step-stare or continuous", "Frequency [kHz] 10-40", "Minimum gate length [m]", "Maximum range [m]" do not tell what settings were actually used.

Fig. 13. Is this relevant information for this study? If not, please remove.

L317 Which SNR limit does this correspond to?

L333-335 Please see Manninen et al. (2016) and Vakkari et al. (2019) for post-processing Halo Stream Line data.

L444 Please check "exposes the rotor to a non-homogeneous flow resulting during off-design operations"

L508-509 "This analysis has also confirmed that the optimal scanning strategy identified by the LiSBOA has been that producing the most accurate flow statistics." This was not shown. Especially, LiSBOA was not compared to any other retrieval from lidar data. Also, on L377-378 the authors state "Instead, an a posteriori analysis of the statistics retrieved is recommended to select the best sigma−m values."

L516-517 "The mean velocity and turbulence intensity extracted 1D upstream of the rotors have agreed well with the values provided by the nacelle anemometers, with maximum discrepancies as low as 3%." On the other hand, Fig. 21 indicates >10% differences in retrieved turbulence intensity depending on the parameters selected for LiSBOA. How well does the upstream comparison represent LiSBOA performance?

L520-521 "Two noticeable advantages of the LiSBOA arise from the present work: first, the LiSBOA allows a straightforward yet effective design of LiDAR scans, which exploits only basic knowledge about the flow under investigation and the LiDARs used." This seems quite optimistic statement to me. The required "knowledge about the flow" is not really basic. For scan design, the integration time per measurement and elevation angle(s) of the PPI scans are not given by LiSBOA. When these are decided, LiSBOA

does optimise the azimuth angle step. Furthermore, the requirement of stationary flow over an extended period (an hour or more) is a serious limitation.

References

Manninen, A. J., O'Connor, E. J., Vakkari, V. and Petäjä, T.: A generalised background correction algorithm for a Halo Doppler lidar and its application to data from Finland, Atmos. Meas. Tech., 9(2), 817–827, doi:10.5194/amt-9-817-2016, 2016.

Vakkari, V., Manninen, A. J., O'Connor, E. J., Schween, J. H., van Zyl, P. G. and Marinou, E.: A novel post-processing algorithm for Halo Doppler lidars, Atmos. Meas. Tech., 12(2), 839–852, doi:10.5194/amt-12-839-2019, 2019.

---

## Author Comment (AC1) · 25 Nov 2020

**Reply to the comments provided by Anonymous Referee #1 on the manuscript amt-2020-228 entitled "LiSBOA: LiDAR Statistical Barnes Objective Analysis for optimal design of LiDAR scans and retrieval of wind statistics. Part II: Applications to  real LiDAR data of wind turbine wakes ", by S. Letizia, L. Zhan and G.V. Iungo**

The authors sincerely thank the referee for the thorough review and the detailed comments. Our replies are reported in the following. References to pages and lines are based on the revised marked-up manuscript.

**Comments:**

*In this study, the authors build on the theoretical work discussed and presented in the companion part 1 paper for reconstructing the wind fields downstream of wind turbines to measure the properties of the turbine wakes. Specifically, the velocity deficit and turbulence intensity are measured. The authors first demonstrate this capability using a virtual lidar simulator to quantify the expected errors, then also demonstrate the capability on measurements near a wind farm in Colorado. The results look compelling, and there is some comparison with in situ measurements to validate the wind field reconstruction  here anemometers were installed. Overall, this is a nice demonstration of the novel technique and the analysis of the wind turbine wakes will be of interest to those in the wind energy field. Still, the virtual lidar simulator needs to be revised as there are several modifications that could be made to it to obtain more realistic results, which will yield a more accurate understanding of how to interpret real-world measurements. This analysis will require significant additional data analysis. Thus, I recommend major revisions to this manuscript after which it may be acceptable for full publication in AMT.*
R: We thank the Reviewer for the positive feedback. The manuscript has been updated to address the comments arisen. It is noteworthy that the virtual LiDAR section has been moved to the companion paper Part I.

a)      *Line 15: It would be helpful to include all the symbols used in the paper in this list, not just those used in LiSBOA.*
R: We have added all the symbols to the nomenclature as suggested by the Reviewer.

b)      *Line 127: This should be projection of the wind vector, not velocity, onto the laser beam to really represent a lidar measurement.*
R: That sentence has been revised (L 507 of Part I): "This method minimizes the turbulence damping while retaining the geometry of the scan and the projection of the wind velocity vector onto the laser beam direction".

c)      *Eq. 1: What is u here? Since there is no arrow over it, I'll assume it is just the streamwise component of the wind within the LES simulator, and not the full 3-D vector. To truly simulate a measurement, it should be the full 3-D wind vector as the radial velocity is not only affected by the streamwise component, but also the vertical and crosswise components (whose means are zero, but instantaneous turbulent perturbations are not). This may have significant effects on the results.*

R: The symbol $\boldsymbol{u}$ **(Bold fonts)** indicates the 3D wind velocity vector. At L 513 of Part I, it is now reported, "… $\boldsymbol{u}$ is the instantaneous velocity vector and the dot indicates scalar product". Also, at L 163 of Part I it is reported: "…(bold symbols indicate vectorial quantities)…".

d)      *Line 157: Why is a freeslip enforced on the bottom of the domain? That does not produce a realistic logarithmic wind profile.*
R: That's correct. For the sake of generality, we used LES data with uniform incoming velocity to avoid typical wake distortion induced by wind shear and providing clearer data analysis. More realistic scenarios are then considered through the LiDAR data presented in Sect. 3 of Part II. At L 453 of Part I, it is now reported "For the sake of generality, a uniform incoming wind is generated by imposing freeslip conditions at the top and bottom of the numerical domain."

e)      *Line 164-166: The text becomes very confusing to this reader around here. The authors should make it clear that the optimal design of the lidar scan is based on the flow characteristics. Thus, the flow characteristics shown and discussed in the next several paragraphs come from the raw LES field. It might be helpful to make the analysis of the LES flow statistics its own subsection to provide clear separation from the lidar simulator itself. It was confusing to me to see lidar simulator results in Fig. 1 immediately followed by analysis of the LES field, before returning to the lidar simulator again. Sect. 2 could benefit from some reorganization as well, to mitigate alternating between the two separate subjects.*
R: We thank the Reviewer for this useful comment. This section has been re-organized by describing in detail the LES dataset and respective statistics first, then presenting the results obtained through the virtual LiDAR and the LiSBOA.

f) *Line 177: Is the integral time scale calculated using a time series of the streamwise velocity in the LES field?*
R: That is correct. At L 467 of Part I, it is now reported: "The integral time-scale is evaluated integrating the sample biased autocorrelation function of the time series of $u$ up to the first zero-crossing (Zieba and Ramza, 2011).

g) *Line 185, Fig 2, Fig. 3: Clarify what is meant by the spectra (and other features) are averaged azimuthally. What does that mean exactly?*
R: By leveraging the wake axisymmetry (see e.g. Iungo et al. 2013), velocity statistics and spectra are averaged azimuthally for the sake of clarity and to increase statistical significance. Specifically, spectra of velocity and turbulence intensity are calculated in the 3D Fourier space, then averaged azimuthally by leveraging the wake axisymmetry and reported as a function of streamwise and radial wavenumbers, $[k_x, k_r]$. Similarly, the velocity statistics calculated at each point of the domain are averaged in the azimuthal direction.

h)  *Line 203: Just to be clear, the constant angular resolution $\Delta\theta$ is for both azimuth and elevation, correct? That is $\Delta\theta = \Delta\beta$.*
R: The Reviewer is right. This is now better clarified in the manuscript at L 544 of Part I.

i)      *Line 225: State the equation for the equivalent velocity approach.*
R: The equation for the streamwise equivalent velocity is now added to the manuscript (Eq. (18)).

*j) Fig. 6/7 (and discussion of it): It would be help to indicate over how much time these statistics are computed over. Based on the statistics, I think it's 160 sec but I may be wrong.*

R: The statistics are calculated over the whole sampling time ($T = 750$ s, see L 454 of Part I). At L 592 of Part I, it is now reported: "The 3D fields of mean velocity and turbulence intensity calculated over $T = 750$ s through first optimal configuration, (i.e. $\Delta\theta = 2.5°, \sigma = 1/4, m = 5$), are rendered in Figs. 13 and 14, respectively".

*k) Sect 2: Doppler wind lidar measurements are subject to error that increases with decreasing SNR; as SNR typically decreases with range, the velocity measurement also becomes less accurate. This error should be considered within the wind lidar simulator for more realistic results of true measurements.*

R: We agree with the Reviewer that the accuracy of the LiDAR measurements is highly dependent on SNR. However, in the present analysis, we only consider quality-controlled LiDAR data, namely the data are initially filtered based on SNR (see e.g Dynamic Filter, Beck and Kühn 2017). Indeed, this analysis aims to assess the capabilities and accuracy of the LiSBOA, without including other sources of error. At L 531 of Part I, it is now reported: "It is noteworthy that the accuracy estimated through the present analysis only includes error due to the sampling in time and space, and data retrieval. Other error sources, such as the accuracy of the instrument (Rye and Hardesty, 1993; O'Connor, 2010), are not included and should be coupled to the LiSBOA estimates for a more general error quantification (Wheeler and Ganji, 2010b)".

*l) Line 350: Clarify how the wind speed variability is corrected by making the LOS velocity non-dimensional, this is not obvious to the reader.*

R: The wind velocity is made non-dimensional by dividing by the wind data by incoming wind velocity from met tower #1. Now, it is clarified in the text at L 209: "Specifically, the wind speed variability is corrected by making the line-of-sight velocity non-dimensional with the incoming wind speed. To this aim, the instantaneous velocity field measured by the LiDAR is divided by the synchronized mean wind speed obtained from the met tower #1, as explained above."

*m) Figure 17: The timestamps above each PPI plot (panels c-e) are confusing and should be removed. It's unclear why each time stamps spans >6 hours.*

R: Thank you for pointing out this mistake. Now the time stamp has been removed.

**Editorial comments:**

*a)    Line 315: Need a space between 65deg and with.*

R: We fixed the typo, thank you.

[revised manuscript text omitted]

---

## Author Comment (AC2)

**Reply to the comments provided by Anonymous Referee #2 on the manuscript amt-2020-228 entitled "LiSBOA: LiDAR Statistical Barnes Objective Analysis for optimal design of LiDAR scans and retrieval of wind statistics. Part II: Applications to  real LiDAR data of wind turbine wakes", by S. Letizia, L. Zhan and G.V. Iungo**

The authors thank the referee for the thorough review and the detailed comments. Our replies are reported in the following. References to pages and lines are based on the revised marked-up manuscript.

**Comments:**

*This manuscript applies the LiSBOA algorithm introduced in the companion paper (Part I) to retrieve wind speed and turbulence intensity from Doppler lidar measurements in wind turbine wakes. A LES data set is used to investigate the quality of LiSBOA algorithm and LiSBOA retrievals from ground-based Doppler lidar measurements are compared to anemometer measurements. This study is within the scope of AMT, but there are some major issues that need to be addressed before it can be accepted. Overall, this manuscript is rather long and has a large number of figures. I suggest to move the LES discussion (Section 2) to Part I (where it can replace the synthetic data) to keep this manuscript focused on the actual measurements.*
R: We appreciate the positive feedback on the assessment of the LiSBOA. We implemented the modifications suggested by the Reviewer to improve the quality of the manuscript. The virtual LiDAR section has been moved to Part I, as recommended.

**Major comments:**

*1.      L160 Some measurements have been conducted with lidars located at the nacelle, but in this study only ground-based lidars are utilised. Please use the LES to assess the quality of ground-based lidar measurements.*
R: The reviewer is right that the experimental data analyzed in this manuscript were collected only through ground-based LiDARs. However, performing the LiSBOA analysis of the LES dataset by simulating a ground-based LiDAR will only imply results and figures more difficult to interpret due to the inclination angle of the PPI measurement planes due to the necessity of limiting the elevation angle, necessary to apply the equivalent velocity approach (Zhan *et al.* 2019, 2020). In the sample figure below, results equivalent to those obtained for the nacelle-mounted LiDAR of Fig. 13-14 for step-stare LiDAR placed 1000 m ($\sim 8\ D$) downstream are reported. The $AE_{95}$ is 4.9% and 5.1% for this optimal configuration, so just slightly higher than the error obtained for nacelle-mounted LiDAR. Such error is nevertheless sensitive to the location of the ground-based LiDAR.  Since the scope of the virtual LiDAR analysis is to assess the accuracy of the LiSBOA for a widely-used wind energy application (e.g. Trujillo et al., 2010, Fuertes et al., 2018, Reiwardt et al., 2020), we would rather keep the analysis of the LES data by simulating a nacelle-mounted LiDAR to generate more compelling and general results for the assessment, as those reported in Figs. 10, 13, 14, 16, and 17. To underscore that the error is case-specific, at L 437 of Part I, it is now reported "The LiSBOA algorithm is applied to a synthetic dataset generated through the virtual LiDAR technique to assess accuracy in the calculation of statistics for a wind turbine wake probed through a scanning LiDAR installed at the turbine nacelle".

[Figure]

3D rendering of mean streamwise velocity and turbulence intensity retrieved through LiBSOA from a virtual step-stare LiDAR dataset installed 1000 m downstream the turbine described in the paper.

*2.       L165-198 Determining the flow characteristics for LiSBOA is not trivial, as seen here and in Figs. 2-3. Please test LiSBOA sensitivity to the flow characteristics. In atmospheric applications they may not be known with good enough accuracy, or they may vary during measurements.*
R: We agree with the Reviewer that the sensitivity of the LiSBOA to the input parameters should be discussed more in detail. A new section has been devoted to this topic, i.e. Sect. 6 in Part I, where the effects of the uncertainty on the flow characteristics is assessed and the robustness of the LiSBOA is discussed.

**Specific comments**

*3.       L17 "angular resolution" which angle?*
R: We corrected the sentence, it is the resolution in the azimuth angle.

*4.       L156 Please state resolution in metres.*
R: Thank you for pointing out this missing information. We expressed the resolution in rotor diameters instead, for the sake of generality (L 451 of Part I).

*5.       L157 What is "A radiative condition is imposed at the outlet"?*
R: We added the reference Orlanski 1976 for the radiative (or open boundary) boundary condition. Also called open boundary condition, it has the following form (Debnath et al., 2017):

$$\frac{\partial u_i}{\partial t} + C_i \frac{\partial u_i}{\partial x_i} = 0, i = 1,2,3$$

Where $C$ is called phase velocity. It has been used in LES studies of turbulent wakes (e.g. Vollmer et al., 2016, Ciri et al, 2017, Debnath et al., 2017).

6.    *L157 "freeslip is enforced on the top and bottom" This doesn't create a realistic wind profile. It may not hamper the use of the LES to validate LiSBOA algorithm but should be justified in the text.*
R: That's correct. For the sake of generality, we used LES data with uniform incoming velocity to avoid typical wake distortion induced by wind shear and providing clearer data analysis. More realistic scenarios are then considered through the LiDAR data presented in Sect. 3, Part II. At L 453 of Part I, it is now reported "For the sake of generality, a uniform incoming wind is generated by imposing freeslip conditions at the top and bottom of the numerical domain."

7.    *Fig. 1. Please state the integration time per radial measurement for panels b-d.*
R: We added more information about the scan parameter in the caption and the text. In the caption of Fig. 10 in Part I, it is now reported: "Snapshot at the hub-height horizontal plane of the wake generated by the 5-MW NREL reference wind turbine: **(a)** LES streamwise velocity; **(b)** ideal virtual LiDAR with angular resolution $\Delta\theta = 2.5°$, zero elevation, accumulation time $\tau_a = 0.5$ s, gate length $\Delta r = 25$ m; **(c)** step-stare virtual LiDAR (same settings); **(d)** continuous mode virtual LiDAR (same settings)."

8.    *L211 Please make it clear that the aliasing here is an issue for LiSBOA retrieval, not for the lidar itself.*
R: That's correct. Correct, however now that part has been moved to Part I, so that the theory and virtual LiDAR descriptions are in the same manuscript, so this discussion should be clearer.

9.    *L215 What is the total sampling time?*
R: The total sampling time is 750 s, which is reported at L 454 and reported again now at L 592 of Part I.

10.    *L226 Please define "equivalent velocity approach".*
R: We added the equation for equivalent velocity as requested (Eq. 18) as suggested.

11.    *L227 "turbulence intensity" could be defined at the first use.*
R: The definition of turbulence intensity is now reported at L 103 of Part I.

12.    *Table 2. Please give the actual operating specifications used in this study. E.g. "Scanning mode Step-stare or continuous", "Frequency [kHz] 10-40", "Minimum gate length [m]", "Maximum range [m]" do not tell what settings were actually used.*
R: The table has been revised as suggested.

13.    *Fig. 13. Is this relevant information for this study? If not, please remove.*
R: The figure is cited at L 207 when discussing the stationarity of the coefficient of thrust for the selected subset of data, so we would prefer to keep it.

*14. L317 Which SNR limit does this correspond to?*
R: The LiDAR data were filtered with the dynamic filter proposed by Beck and Kühn, 2018, which uses "normalized SNR and radial velocity" and a probabilistic approach to select valid data. Furthermore, we added an additional threshold of -18 dB for the SNR, which is the recommended value for Halo (Manninen et al., 2016).

*15. L333-335 Please see Manninen et al. (2016) and Vakkari et al. (2019) for postprocessing Halo Stream Line data.*
R: We thank you the Reviewer for suggesting these references, which are already familiar to us. In these papers, a correction method to discriminate the LiDAR signal between instrumental noise, cloud and aerosol signal is proposed. Our data were collected at lower heights and much higher SNR and for a range lower than 1300 m, for which the dynamic filter proposed by Beck and Kühn, 2018 performs satisfactorily.

*16. L444 Please check "exposes the rotor to a non-homogeneous flow resulting during off-design operations"*
R: Thank you for spotting this error, now it has been fixed. At L 305, it is now reported: "exposes the rotor to a non-homogeneous flow resulting in a severely off-design operation".

*17. L508-509 "This analysis has also confirmed that the optimal scanning strategy identified by the LiSBOA has been that producing the most accurate flow statistics." This was not shown. Especially, LiSBOA was not compared to any other retrieval from lidar data. Also, on L377-378 the authors state "Instead, an a posteriori analysis of the statistics retrieved is recommended to select the best sigma-m values."*
R: The Reviewer is right, and these statements have been removed.

*18. L516-517 "The mean velocity and turbulence intensity extracted 1D upstream of the rotors have agreed well with the values provided by the nacelle anemometers, with maximum discrepancies as low as 3%." On the other hand, Fig. 21 indicates >10% differences in retrieved turbulence intensity depending on the parameters selected for LiSBOA. How well does the upstream comparison represent LiSBOA performance?*
R: The error is expressed as a percentage of the incoming wind speed for velocity and in absolute terms for the turbulence intensity, because TI is already a non-dimensional quantity (e.g. TI=10% and TI=12% would have a 2% error). The agreement with the SCADA anemometer (i.e. upstream flow of the individual turbine) is a good parameter to assess the performance of the LiSBOA, since the flow under investigation is perturbed by wakes, except for the leading turbine F01. The most challenging quantity to reproduce is the turbulence intensity, which is indeed estimated satisfactorily by the LiSBOA. To clarify better this aspect, at L 377it is now reported: "The mean velocity and turbulence intensity extracted 1D upstream of the rotors have agreed well with the values provided by the nacelle anemometers, with maximum discrepancies as low as 3% of the undisturbed wind speed for the mean velocity and 3% (in absolute terms) for the turbulence intensity". In the abstract as well, a L 12: "Maximum discrepancies as low as 3% for the mean velocity (with respect to the freestream velocity) and turbulence intensity (in absolute terms) endorse the application of the LiSBOA for LiDAR-based wind resource assessment and diagnostic surveys for wind farms."

*19.    L520-521 "Two noticeable advantages of the LiSBOA arise from the present work: first, the LiSBOA allows a straightforward yet effective design of LiDAR scans, which exploits only basic knowledge about the flow under investigation and the LiDARs used." This seems quite optimistic statement to me. The required "knowledge about the flow" is not really basic. For scan design, the integration time per measurement and elevation angle(s) of the PPI scans are not given by LiSBOA. When these are decided, LiSBOA does optimise the azimuth angle step. Furthermore, the requirement of stationary flow over an extended period (an hour or more) is a serious limitation.*

R: We understand the comment and we toned down the statement that now reads (L 382): "Two noticeable advantages of the LiSBOA arise from the present work: first, once the wavelengths of interest and the LiDAR basic setup are selected, the LiSBOA allows a systematic and effective design of LiDAR scans, which includes all the essential information of the flow under investigation". Please refer to the newly added Sect. 6 in part I for a discussion about the sensitivity of the algorithm to the inputs and the issues connected with non-stationarity.

[revised manuscript text omitted]

---

## Author Response (AR2)

**Reply to the comments provided by the Anonymous Referee #2 on the manuscript amt-2020-228 entitled "LiSBOA: LiDAR Statistical Barnes Objective Analysis for optimal design of LiDAR scans and retrieval of wind statistics. Part II: Applications to LiDAR measurements of wind turbine wakes", by S. Letizia, L. Zhan and G.V. Iungo**

The authors thank the referee for the further comments. Our replies are reported in the following. References to pages and lines are based on the revised marked-up manuscript.

**Comments**

*I would like to thank the authors for their reply. I have only one minor comment. At several places in the manuscript it is stated that LiSBOA has been used to select or optimize the scanning strategy, while it seems to be used only to adjust the azimuth angle resolution and most of the scan settings have been decided without using LiSBOA. Please state clearly which part of the scan strategy was designed with LiSBOA. Some statements to consider:*
*L1-2 "The LiDAR Statistical Barnes Objective Analysis (LiSBOA), presented in Letizia et al. (2020), is a procedure for the optimal design of LiDAR scans ..."*
*L57-58 "... the data collection strategy is optimally designed through the LiSBOA."*
*L100-103 "The scope of this study is dual: first, assessing the capabilities provided by the LiSBOA for the optimal design of the LiDAR scanning strategy by maximizing the statistical accuracy of the measurements and coverage of the sampling domain with the prescribed spatial resolution;"*
*L156-157 "The present section aims to explore the potential of the LiSBOA for the optimal design of a LiDAR experiment, data postprocessing, and reconstruction of 3D flow statistics."*
*L357 "The optimal LiDAR scanning strategy has been selected through the LiSBOA, ..."*
*L366-368 "Two noticeable advantages of the LiSBOA arise from the present work: first, once the wavelengths of interest and the LiDAR basic setup are selected, the LiSBOA allows a systematic and effective design of LiDAR scans, which includes all the essential information of the flow under investigation and the LiDARs used."*

R: We now clarify that in this paper the LiSBOA is used only for the optimization of azimuth resolution, being the other parameters fixed by the atmospheric conditions and the geometry of the scan. For instance:
At L 4 it is now reported: "For both case studies, the LiSBOA is leveraged for the optimization of the azimuthal step of the LiDAR and the retrieval of mean equivalent velocity and turbulence intensity fields".

At L 57 it is now reported: "The LiSBOA also performs adequate filtering of small-scale variability in the mean velocity field and mitigation of the dispersive stresses on the higher-order statistics provided that the algorithm is tuned based on the characteristics of the flow under investigation and the free parameters of the LiDAR scan are optimally designed through the LiSBOA".

At L 102 it is now reported: "The scope of this study is dual: first, assessing the capabilities provided by the LiSBOA for the optimal selection of the angular step of the LiDAR scans by maximizing the statistical accuracy of the measurements and coverage of the sampling domain with the prescribed spatial resolution…".

At L 158 it is now reported: "The present section aims to explore the potential of the LiSBOA for the selection of the optimal azimuthal resolution of a LiDAR scan, data post-processing, and reconstruction of 3D flow statistics."

At L 355 it is now reported: "The LiDAR Statistical Barnes Objective Analysis (LiSBOA) has been applied to two different cases of wind turbine wakes to estimate the optimal azimuthal step of the LiDAR and retrieve mean velocity and turbulence intensity fields".

At L 360 it is now reported: "The optimal azimuthal resolution of the LiDAR scan has been selected through the LiSBOA, while the mean velocity and turbulence intensity fields retrieved through the LiSBOA have offered a detailed insight of the wake morphology".

At L 370 it is now reported: "Two noticeable advantages of the LiSBOA arise from the present work: first, once the wavelengths of interest and the LiDAR basic scanning parameter dictated by the atmospheric conditions and target position are selected, the LiSBOA allows a systematic and effective optimization of the azimuth resolution, which includes all the essential information of the flow under investigation and the LiDARs used".